# Flower-shaped 2D crystals grown in curved fluid vesicle membranes

Hao Wan [1], Geunwoong Jeon [2], Weiyue Xin [3], Gregory M. Grason [1] & Maria M. Santore [1] ✉

The morphologies of two-dimensional (2D) crystals, nucleated, grown, and integrated within 2D elastic fluids, for instance in giant vesicle membranes, are dictated by an interplay of mechanics, permeability, and thermal contraction. Mitigation of solid strain drives the formation of crystals with vanishing Gaussian curvature (i.e., developable domain shapes) and, correspondingly, enhanced Gaussian curvature in the surrounding 2D fluid. However, upon cooling to grow the crystals, large vesicles sustain greater inflation and tension because their small area-to-volume ratio slows water permeation. As a result, more elaborate shapes, for instance, flowers with bendable but inextensible petals, form on large vesicles despite their more gradual curvature, while small vesicles harbor compact planar crystals. This size dependence runs counter to the known cumulative growth of strain energy of 2D colloidal crystals on rigid spherical templates. This interplay of intra-membrane mechanics and processing points to the scalable production of flexible molecular crystals of controllable complex shape.

The importance of ultrathin and flexible materials motivates a focus on nanometrically thin bendable crystals whose sheet-like character and large lateral areas qualify them as 2D solids. Experiments, simulations, and theory show that a 2D crystal growing on a fixed spherical template avoids topological defects via a stress-triggered boundary instability[1,2], producing a progression from compact, hexagonal domains to highly anisotropic protrusions and stripes which reduce the energetic cost of Gaussian curvature[3–6]. The characteristic domain length scale is controlled by the interplay between in-plane stresses, line energy, and sphere radius[3,4,7,8]. Thus on smaller templates with greater curvature, the transition from compact to striped/protruding morphology occurs earlier during crystal growth, with narrower protrusions and stripes emanating from smaller compact domain cores.

Here we show how the morphologies of 2D crystals growing within a flexible elastic 2D fluid having a closed topology, for instance, solid domains integrated into the fluid membrane of a giant unilamellar vesicle shown schematically in Fig. 1a, are controlled by a fundamentally distinct mechanism. The in-plane solid elasticity of 2D crystals, i.e., a preference for flatness or cylindrical bending[9,10], favors solid domains with zero Gaussian curvature. Because topology requires that the total Gaussian curvature of the composite vesicle is constant, the expulsion of Gaussian curvature from the solid domain must be redistributed to the 2D fluid[11]. This in turn produces a complex interplay between the shape and morphology of the crystal and the system's global shape and bending energy. We employ fluid $L_\alpha$ phase phospholipid bilayer membranes[12,13] containing integrated solid membrane domains as a platform to explore how 2D crystals, growing within a curved 2D elastic fluid, adjust both their morphology and curvature to minimize the total energy. The size sensitivity of the resulting morphological instability runs counter to that for crystallization on rigid spherical templates[3,4,7]. Further, thermal membrane contractions and water permeation from the vesicle tune membrane tension and energy to scalably produce a vesicle size dependence of crystal morphologies. When thermal history and osmotic preconditioning are fixed for the entire suspension, more elaborate

[1]Department of Polymer Science and Engineering, University of Massachusetts, 120 Governors Drive, Amherst, MA 01003, USA. [2]Department of Physics, University of Massachusetts, 710 N. Pleasant Street, Amherst, MA 01003, USA. [3]Department of Chemical Engineering, University of Massachusetts, 686 N. Pleasant Street, Amherst, MA 01003, USA. ✉e-mail: santore@mail.pse.umass.edu

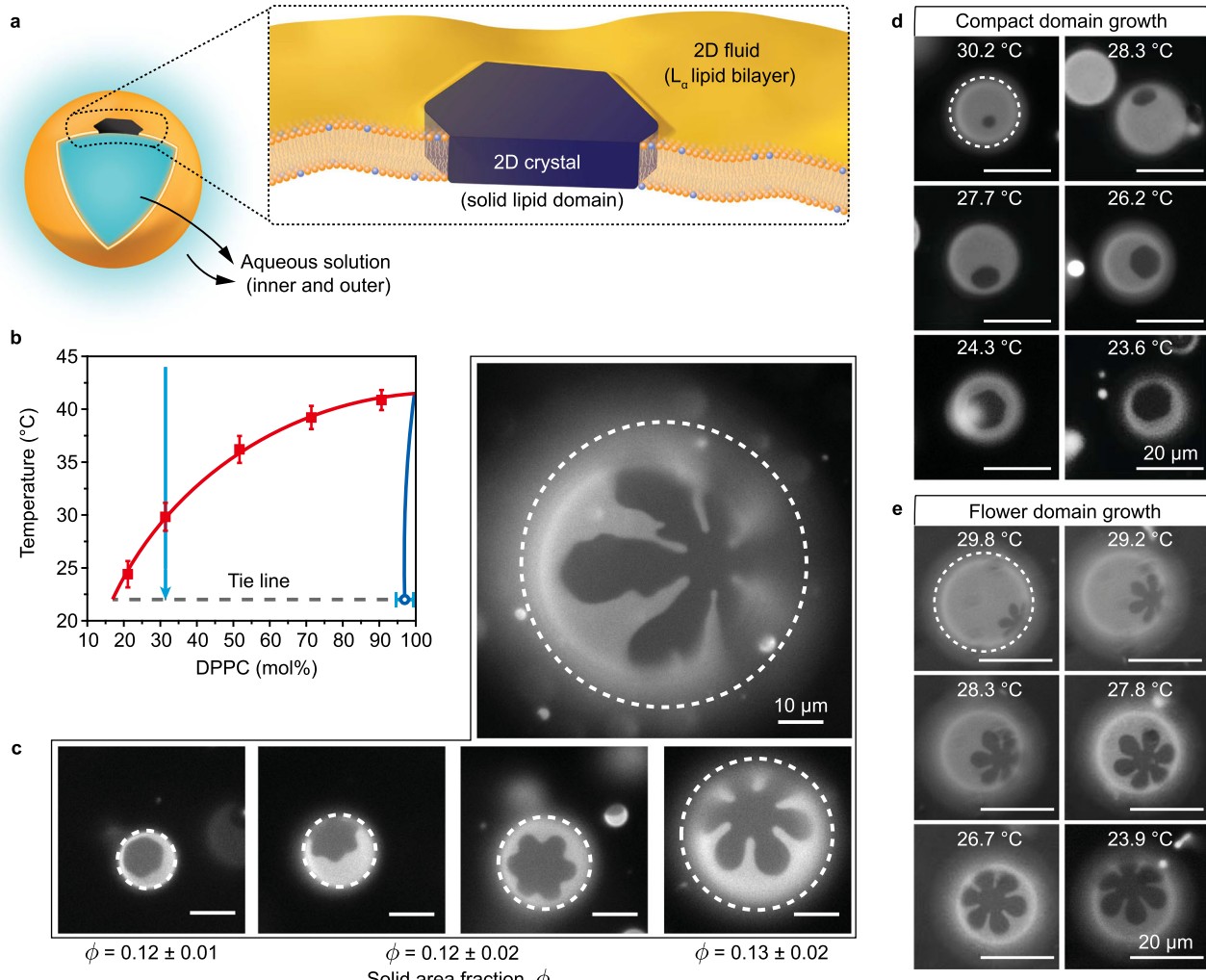

**Fig. 1 | Solid crystals in fluid vesicle membranes. a** Schematic of a solid crystal (dark) in a 2-D fluid (yellow) comprising a $L_\alpha$ lipid bilayer. **b** Phase diagram and cooling trajectory for giant unilamellar vesicle membranes of lipid mixtures of DPPC and DOPC and <0.1 mol% tracer. $L_\alpha$ phase boundary (red squares) based on the first appearance of solid domains upon cooling at each composition. Solid datum (blue circle) approximates solid domains known to be nearly pure in DPPC[54,55]. Error bars are standard deviation and include 6 independent measurements for each composition. **c** Examples of DPPC solid crystal shapes seen on vesicles of different sizes, all in the same batch and shown with the same 10-μm scale bar. The solid crystalline domains are dark shapes, while the $L_\alpha$ fluid is bright due to the fluorescent tracer. Solid area fractions for different shape types (convex hexagon, ninja star flower, simple flower) are included and represent averages of 15-20 vesicles of each shape type, detailed in Suppl. Note 2. The dashed boundaries indicate the vesicle size, determined by a separate image taken a few moments apart focusing on the equatorial plane rather than the crystal-containing surface. Scale bars are 10 μm. **d, e** Two series of vesicle images, each showing a progression for a single vesicle and its growing crystal shape during cooling. The first image in each series was acquired as soon as a crystal could be identified and focused. The rest of the images continue to room temperature. The two vesicles differ only in size. The dashed boundaries in each first image indicate the vesicle size, based on a separate image recorded at a slightly different time, focusing on the equatorial plane rather than the pattern. Scale bars are 20 μm.

crystal shapes grow systematically on larger more gradually curved vesicles, at the same membrane composition. The favorable comparison between experiments with phospholipid vesicles and the continuum model, which imposes no constraints on the molecular makeup of fluid and solid domains, argues for the generality of the mechanism proposed here.

## Results

### Emergent shapes of crystallized domains

In giant unilamellar vesicle membranes containing two or more phospholipids, ordered crystalline membrane domains, some with intricate shapes[14], can coexist within the fluid membrane phase, called a $L_\alpha$ phase[15–18]. In this work the $L_\alpha$ fluid membrane contains a mixture of 1,2-dipalmitoyl-*sn*-glycero-3-phosphocholine (DPPC) and 1,2-dioleoyl-*sn*-glycero-3-phosphocholine (DOPC), plus a fluorescent tracer lipid (1,2-dioleoyl-*sn*-glycero-3-phosphoethanolamine-*N*-(lissamine

rhodamine B sulfonyl) (ammonium salt), Rh-DOPE) that is excluded from the nearly pure ordered DPPC solid domains[16,19–21], enabling their visualization in membranes containing both fluid and solid domains. The $L_\alpha$ fluid membrane can bend and stretch elastically and, because it can also shear freely[22–25], it assumes Gaussian curvature at a low cost. Unlike the fluid membrane, solid phospholipid domains possess identical crystalline order in both leaflets[26–30] and correspondingly non-zero 2D shear moduli[22], imparting shear rigidity that limits access to shapes like spheres that have non-zero Gaussian curvature. Solid shear elasticity underlies the geometrically nonlinear coupling between Gaussian curvature and in-plane strain of 2D sheets[9]. While changes in mean curvature (i.e., bending) incur modest cost due to the nanometrically thin dimensions[22,25,31–34], non-zero Gaussian curvature in the solid leads to prohibitively large shear strains that grow with the lateral dimensions of the 2D crystal[7,35]. Notably, the formation of topological defects that form in some curved 2D crystals[36–38]

can only relax a fraction, but not all, of the large thermodynamic costs of Gaussian curvature[39].

Starting with an overall membrane composition of 30/70 DPPC/DOPC molar ratio in the one phase region of the phase diagram in Fig. 1b, cooling at an appropriate rate produces solid domains, one per vesicle, that first appear just below 32 °C and grow progressively until at room temperature, they occupy area fraction $\phi = 0.12 - 0.14$ of the membrane, according to the room temperature tie line (Fig. 1b). An example mass balance-based calculation of the solid area from the tie line can be found in the Suppl. Note 1. A variety of solid shapes, all with 6-fold symmetry from convex hexagons and highly non-convex stars to flowers, are found with examples shown in Fig. 1c. The room temperature solid area fractions measured from image analysis of 15–20 vesicles of each crystal shape type, summarized within Fig. 1c and detailed in the Suppl. Note 2, is independent of domain shape and agrees with the mass balance on the room temperature tie line. Thus, despite the varied morphologies of the solid domains, all the vesicle membranes adhere to the thermodynamic phase diagram, suggesting mechanisms other than none-equilibrium thermodynamics to explain the varied domain morphology. Important to note, that the composition 30/70 DPPC/DOPC was chosen because it produces only one crystal per vesicle, and enables the beautiful shapes in Fig. 1 to grow in completely without interference by other crystals. Greater DPPC content produces multiple nuclei per vesicle[40], with growing domains interacting and covering a greater area fraction and interfering with the morphologies developed with isolated crystals.

Two unusual features are observed in these systems. First, as shown in two typical examples in Fig. 1d and e, the initially discernable solid shape is preserved as domains grow. This observation suggests crystal growth and shape development by molecular addition at domain edges as opposed to aggregation of small domains to produce large ones. There is also a lack of classical dendritic growth instabilities since the shapes do not branch as they grow. Indeed, the images of Fig. 1d and e are consistent with single crystal growth. Second, evident in Fig. 1c, there is a strong correlation between domain shape and vesicle size. More compact, convex domains are found on smaller vesicles, and solid domains with more extended protrusions and elaborate flower shapes on the larger vesicles. This is particularly remarkable because all these vesicles have the same composition, and same solid area fraction at room temperature, and were processed together to produce nucleation and growth in a single chamber with the same osmotic handling and thermal program. The particular vesicles in Fig. 1c were selectively visualized by translating the microscope stage to focus on different vesicles.

## Vesicle size selects crystal shape

The solid domain shapes comprise a continuum defined by the ratio of the circumradius to inradius, i.e., $\alpha = D_{outer}/D_{core}$. The nominal shape types, discernable by the eye, in Fig. 2b provide a convenient framework for classifying solid domains and always exhibit the $\alpha$ values in Fig. 2b. For instance, $\alpha$ varies from 1–1.15 for hexagons and convex domains that are less sharply faceted, up to >3.5 for serrated flowers. The categories of hexagons, ninja star flowers, simple flowers, and serrated flowers enabled us to establish, in Fig. 2c the dependence of domain shape on vesicle size for over 330 vesicles in three separate runs, all with the same lipid composition, osmotic conditions and thermal history. Figure 2c establishes distinct vesicle size ranges where crystals of different shapes are found. The correlation is strong, with serrated flowers found only on the largest vesicles and for instance, compact hexagonal domains never seen on vesicles greater than 25 μm in diameter.

Counter to expectations based on the literature[3,4,7], the most compact shapes are found on the small vesicles, i.e., those with the greatest curvature, while non-convex flowers are seen on larger vesicles,

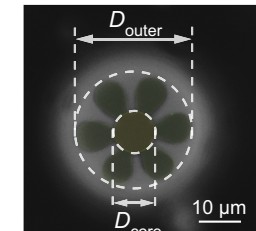

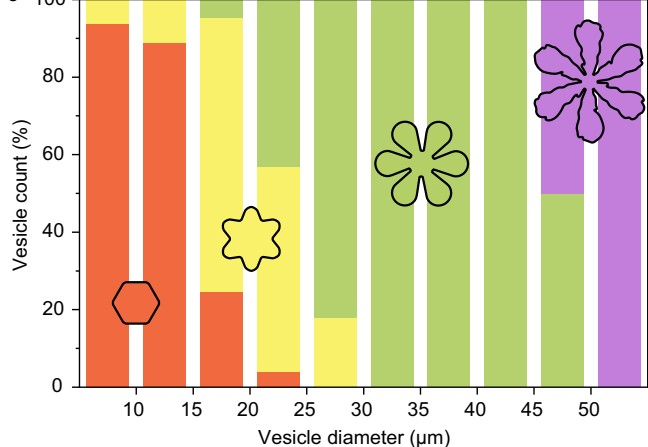

| α | | Shape |
|---|---|---|
| 1-1.15 | ⬡ | Hexagon |
| 1.2-1.8 | ✦ | Ninja star flower |
| 1.8-3.5 | ✳ | Simple flower |
| >3.5 | ✳ | Serrated flower |

**Fig. 2 | Vesicle size dependence of crystal shape. a** Schematic illustrating how α was measured and **b** the range of α for different types of solid shapes. **c** Summary of solid domain shapes found on 330 vesicles of different sizes, including hexagon (red), ninja star flower (yellow), simple flower (green), and serrated flower (purple). More than 100 vesicles were analyzed in each of three separate batches.

having smaller curvature. This correlation runs counter to predictions of Föppl-von Kármán theory[3,7], phase-field crystal modeling[2,5], and experiments of colloidal crystallization on fixed spherical templates[4] that show the threshold size for the transition from compact to anisotropic (e.g., stripe, branched or non-convex) domain shape increases with larger sphere radius.

## Solid mechanics favors developable crystal shapes

To build towards an understanding of the vesicle size-based selectivity of crystal morphology, we start by considering the energetics of a vesicle having total area ($A$) and enclosed volume ($V$) where the fluid membrane contains a solid domain of fixed area fraction ($\phi$). While flexible vesicles need not be spherical, the area-integrated Gaussian curvature is constant, and the standard (Helfrich model) bending energy of fluid membranes favors uniformly spherical vesicles[12], both of which are frustrated by the presence of solid domain. To see this, we first estimate the energetics of forcing the solid crystal domain to conform to a uniform spherical radius $R = \sqrt{A/(4\pi)}$. This energy is composed of the elastic strain energy imposed by the Gaussian curvature and the out-of-plane (i.e., mean curvature) bending energy cost. The former is proportional to $E(strain) \sim YR^2$, where $Y$ is the 2D Young's modulus of the solid, while the latter is proportional to the bending modulus $E(bend) \sim B$[35]. For simplicity, bending modulus $B$ is taken to be the same for the fluid and solid phases. Since the modulus ratio $\sqrt{B/Y} = t$ gives a length scale, $t$, characterizing the elastic thickness[9], and is thus of order of the nanometric lamella thickness, we expect the ratio $E(bend))/E(strain) \sim (t/R)^2 \ll 1$. This implies that vesicles are likely to adopt large, mean-curvature bending deformations without imposing Gaussian curvature on solid domains. In other words, elasticity drives solid domains to take the form of zero-Gaussian developable surfaces[9,41] either remaining planar or else bending cylindrically or in a locally conical geometry, while the $L_\alpha$ fluid phase

will deform to accommodate all the non-zero Gaussian curvature of the closed vesicle.

These mechanics are borne out in studies focusing on the shape of tracer-containing fluid membranes and, to the extent they could be visualized, the solid domains. Creating a challenge for imaging solid domain curvature, membrane tracer lipids, established in other works to integrate into certain solid phospholipid domains[14,16,17], were excluded from the particular crystals in this study. This is shown in Supplementary Fig. 1 for lissamine rhodamine B-1,2-dipalmitoyl-*sn*-glycero-3-phosphoethanolamine, Rh-DPPE, and dioctadecyl tetra-methylindocarbocyanine perchlorate, C18-DiI. Therefore, phase contrast microscopy coupled with epifluorescence was employed to examine the shapes of equatorial sections when the crystals of solid domains were appropriately oriented to reveal their curvature or flatness. Schematics, derived from the surface evolver modeling in the next section, are included simply to guide the eye in interpreting the shapes of tilted crystals.

Figure 3 reveals the curvature of the fluorescently labeled $L_\alpha$ fluid membrane and, in general, suggests primarily flat or cylindrically-curved solid domains. For instance, in Fig. 3a, the $L_\alpha$ fluid phase curves throughout while the compact hexagonal inclusion is flat. The star-shaped domain in Fig. 3b appears to be substantially flat towards its core but at the viewing angle, a star tip is seen to bend isometrically around the vesicle. In Fig. 3c, the flatness across the narrow direction of 3 petals is clear, translating to cylindrical petal curvature in the wrapping direction, still with zero Gaussian curvature.

The behavior of a planar core and largely cylindrical bending of the petals, is strikingly similar to so-called "capillary origami" deformations of solid sheets on liquid drops[42,43]. This suggests that the formation of highly non-convex and multi-lobed solid domains has a

similar effect in vesicles, to enhance the ability to conform closely to spherical shapes without in-plane strain. Unlike the capillary origami scenario, however, the energetics of the overall shape and morphology dependence are controlled by the overall bending energy of the fluid and solid phases.

### Energetics of inflation versus crystal shape

We model the detailed distribution of elastic energy using Surface Evolver calculations[44] of closed vesicles having a single elastic solid (crystal) domain of area fraction, $\phi$, and a fixed dimensionless volume $\bar{v} = \sqrt{36\pi}V/A^{3/2} \leq 1$. $\bar{v}$ is a dimensionless measure of inflation, the actual volume normalized by the volume enclosed by an equal-area sphere. We consider nearly inextensible conditions $t/R = 8.3 \times 10^{-5}$, justified by the nanometric thickness of multi-micron vesicles. For the solid domain, we consider a range of flower-like shapes (detailed in the Methods section) which span from $\alpha = 1$ for circular solid domains to $\alpha > 4$ for large petal morphologies. Notably, some measure of 6-fold symmetry is apparent in nearly all solid domains, whether compact or non-convex in shape. We understand this 6-fold symmetry to derive from an anisotropy of the line-energy of the solid, reflecting the long-range rotational symmetry of the molecular packing, hence giving rise to at least some measure of 6-fold faceting as derived from standard considerations of 2D crystals (i.e., Wulff shape). Given this fixed 6-fold symmetry, the model considers how variable convexity and elaboration of petalled shapes control the elastic energy of the composite vesicle at fixed solid fraction and inflation. That is, in the present work we consider a family of fixed, 6-fold symmetry solid domain shapes in our model and neglect the effect of the anisotropic energy that gives rise to it in experiments. In the absence of a molecular field that biases anisotropic line energy, it is likely that the *n*-fold of a non-convex

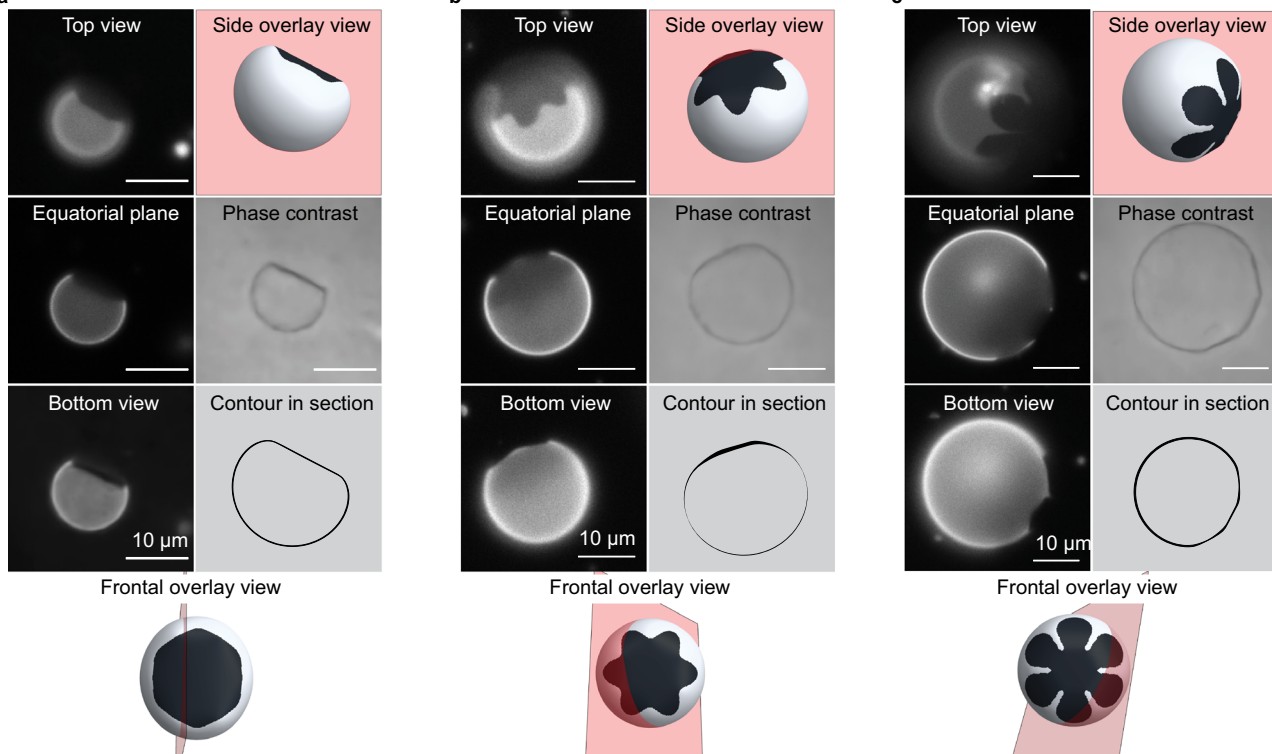

**Fig. 3 | Vesicle shape and solid shape.** Micrographs showing the relationship between domain bending (or lack of it) for typical solid domain shapes: **a** Hexagonal domain **b** Ninja-star flower and **c** Simple flower-shaped domain. Fluorescence micrographs show different focus planes for top, equatorial, and bottom views. An equatorial view in phase contrast is provided in each panel. All the schemes, generated using Surface Evolver, illustrate the position of the solid on the rotated vesicle, the equatorial contour for each case (for comparison to the phase contrast image), and a top view showing the section (red plane) achieved in the micrographs. The phase contrast images have had noise removed to better reveal their boundaries and the original images appear in Supplementary Fig. 2.

domain shape may vary with vesicle inflation and solid domain size, a possibility to be explored elsewhere. We further note that Surface Evolver calculations describe only the thermodynamics of solid domain shape on vesicles of variable inflation, and do not attempt to capture the complexities of dynamic solid domain growth.

We first show elastic energy ground states in Fig. 4a, for slightly inflated conditions (i.e., $\bar{v}$ above the minimal elastic energy state for the starred points in Fig. 4b), mapping the distribution of both mean and Gaussian curvature over vesicle surfaces, resembling those in experiments. In all cases, it is found that the Gaussian curvature is expelled from the solid domains, confirming that their shapes closely follow developable surfaces, i.e., surfaces with everywhere zero Gaussian curvature. Second, we note for compact solid domains on inflated vesicles, large bending (mean curvature) is heavily concentrated in the fluid adjacent to the solid, to accommodate the transition from a mostly spherical fluid matrix and planar solid. Careful inspection of the pattern of mean-curvature in the solid shows that outer regions of the solid are folded along straight lines (known as the generators of developable shape), that do not intersect or end in the solid domain (see Supplementary Fig. 3), a necessary condition for avoiding Gaussian curvature[9,41]. For $\alpha > 1$, we observe that developable folding tends to concentrate at the bases of the petals, resulting in an effective planar flower core and, overall, a more uniform and energetically favorable

mean-curvature distribution over the entire vesicle. Notably, the developable folding of the outer portions of solid domain shapes tends to focus Gaussian curvature in the fluid domain just outside the solid domain, at the apparent intersection of cylindrical folding directions (i.e., generators). The focused Gaussian curvature between the petals in real vesicles is evident in the phase contrast images in Fig. 3c, where the perimeter appears to have corners which in fact occur in the fluid membrane phase. Interestingly, the mean curvature distributions in Fig. 4a suggest that the optimal patterns of developable "folding" of the solid domain break the 6-fold symmetry of the solid domain shape. While it remains to be understood what governs the optimal patterns of internal folds, it is expected the discretization effects at the edges of the curved solid domains introduce the largest source of symmetry-breaking noise in the Surface Evolver. This leads to the selection among many, presumably degenerate, patterns.

Figure 4b shows the reduced energy of the fluid-solid vesicles as a function of inflation, $\bar{v}$. In general, all domain shapes show a non-monotonic trend in energy, with a minimum at a particular inflation, which we denote as $\bar{v}_0$, that becomes deeper over a series of solid shapes with increasingly larger petals (increasing $\alpha$). As shown in the Methods, the membrane tension $\tau$ is proportional to the slope of elastic energy versus $\bar{v}$. Hence the points $\bar{v}_0$ correspond to vanishing tension. The points $\bar{v} > \bar{v}_0$ correspond to tensed and inflated shapes, the

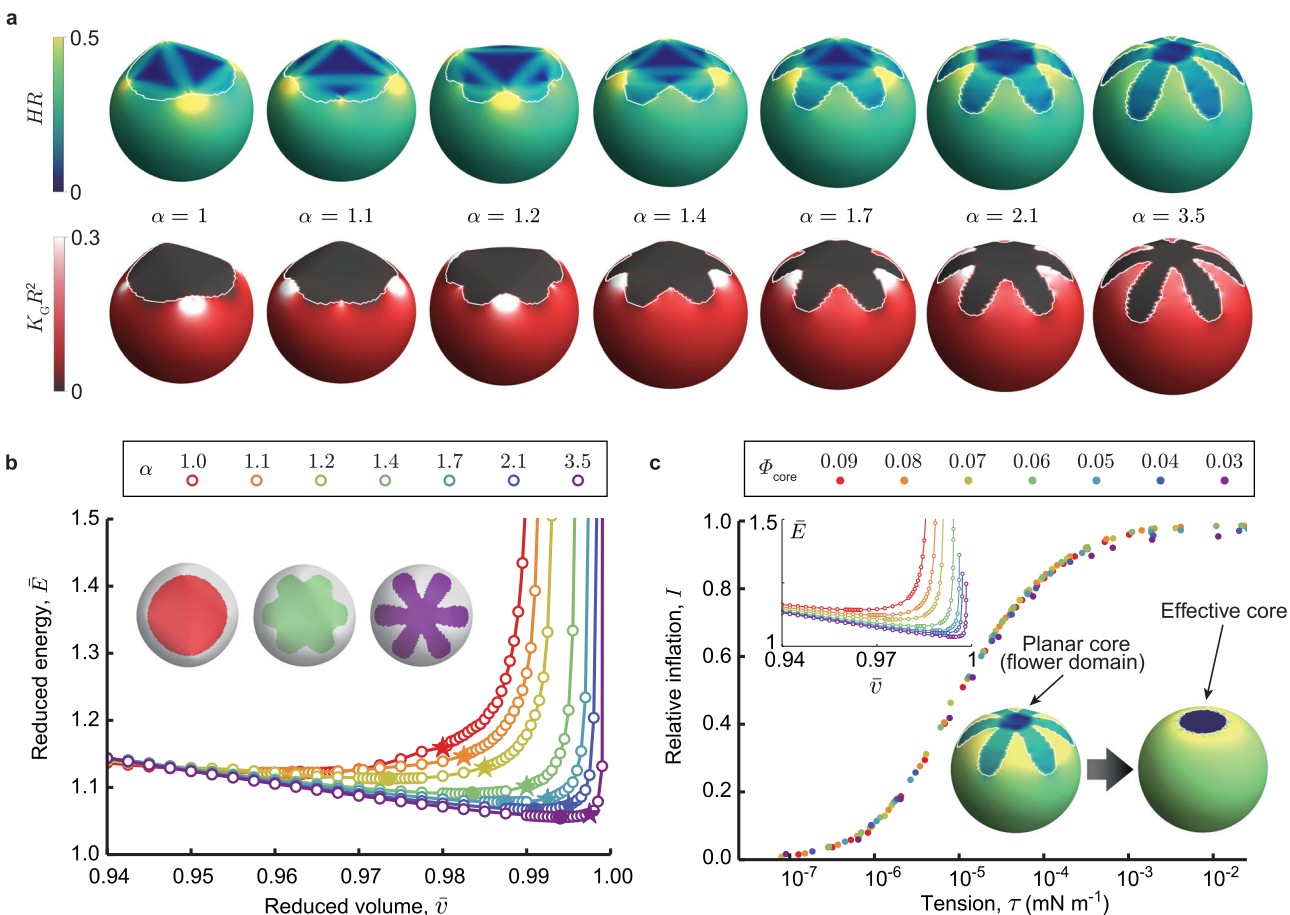

**Fig. 4 | Solid shapes and vesicle energies. a** Simulated shapes of composite vesicles of solid area fraction ($\phi = 14\%$) with flower shapes of various petal/core ratios, showing mean ($H$) and Gaussian ($K_G$) curvature distributions. **b** Reduced elastic energy $\bar{E}$ (normalized by the bending energy of perfectly uniform vesicle, see detailed definition in Methods) of simulated vesicles versus reduced volume $\bar{v}$ for the series of flower shapes in **a**. The tension-free states are highlighted filled points on each curve, while the stars show the slightly inflated volumes used for the examples in **a**. **c** Comparison of an "effective core" model, in which the effect of the

planar core in the flexible, flower-shaped solid is replaced by a rigid disk whose size is chosen to produce matching of $\bar{E}$ in the tension-free states for both models. The inset shows the energetics vs. reduced volume for the sequence of effective core values matching variable flower shapes in **b**. Replotting in terms of the $I$ relative inflation (compared to maximal volume isoperimetric shapes) of effective core shapes shows a generic dependence of relative inflation on tension (here calculated for $R = 10$ μm shapes and taking $B = 25$ $k_B T$).

regime relevant to solid domain formation, as we describe below. For a fixed solid area fraction, as is the case in experiments and computation, the elastic energy decreases with increasing degree of flowering ($\alpha$), and $\bar{v}_0$ shifts to higher inflation. Overall, this shows that the larger the petal-to-core ratio the more uniform the overall mean curvature distribution that can be achieved by a combination of the developable solid shape in the fluid matrix.

Figure 4c shows that the elastic energy versus inflation behavior of the more elaborately shaped flower domains over the range $\alpha \geq 1$ can be mapped onto a much simpler effective core model. In the effective core model, the elasticity effect of the solid domain is modeled by a rigid planar disk with an area fraction $\Phi_{core}$, while the remaining solid is described by bending elasticity (mean curvature) only, as the longer petals are better able to accommodate more nearly spherical vesicles. The inset shows that the appropriate choice of effective core size mimics the predominant trends of the elastic energy dependence of more elaborate shapes in Fig. 4a, b on reduced volumes. This includes the non-monotonic form of elastic energy, and the variable energy and inflation of zero-tension ($\bar{v}_0$) states, as shown in Supplementary Fig. 4. Notably, as shown in Supplementary Fig. 5 the effective core radius closely follows the variation of inner radii of flowers for the sequence $\alpha$ in Fig. 4b. The effective core model provides a simple interpretation for the divergence of elastic energy at large inflation, with more and more mean curvature (bending) cost concentrated at the edge of the planar core with increasing inflation. As the size of the effective core is reduced, the elastic bending cost imposed by the strain-free solid is also reduced, consistent with the decrease of reduced energy with large petal to core ratios in Fig. 4b. A rigid circular core imposes a maximum inflation $\bar{v}^{max}$ shape, a spherical bulb joined to a planar core at finite angle leading to diverging mean-curvature at the core edge. Using this maximal inflation we define $I = (\bar{v} - \bar{v}_0)/(\bar{v}^{max} - \bar{v}_0)$ as a measure of relative inflation, compared to the zero tension state at $\bar{v}_0$, and in Fig. 4c analyze the relationship between relative inflation and tension, $\tau$, predicted by the effective core model. Notably, for all effective core sizes and a given $R$ ( = 10 μm in this example), relative inflation follows the same dependence on $\tau$, switching from low- to high-relative inflation (where elastic energies become significant) at a characteristic tension scale of $10^{-4}$ mN m$^{-1}$. As we describe below, the regulation of inflation by tension is critical to the emergence of non-convex solid domain morphologies.

While the elastic energy is decreased in vesicles whose solid domains contain longer petals that enable the solid to conform more uniformly to curved quasi-spherical shapes (as shown in Fig. 4b), increasingly elaborate flowers with greater fluid-solid perimeters incur an increasing cost of line energy. Comparing the line tension, $\sigma$, to the elastic cost of bending introduces a length scale $B/\sigma$ which is estimated to be in the range of 25 nm based on anticipated values of intra-vesicle fluid/solid domain boundaries. Figure 5 shows the resulting thermodynamically preferred domain shape as a function of vesicle size and inflation. As line energy generically favors compact domains and, as the elastic energy differences for different petal ratios are relatively small at low inflation, compact domains are favored at low inflation or smaller $\bar{v}$. In contrast, as inflation increases to values approaching spherical shapes $\bar{v} \to 1$, the elastic bending costs far exceed the additional line energies of petals, stabilizing petal formation. Indeed, once full inflation is approached, the preference for flower shapes with large petals sets in abruptly and small changes in variables shift the preference for one shape flower over another. These results suggest that the predominant factor controlling flower stability and degree of petal to core area distribution is the degree of vesicle inflation, as captured by $\bar{v}$. Experimental data are included in Fig. 5 for 2 different cooling rates, and show good qualitative agreement with modeling, most notably that the observed degree of flowering is strongly correlated with apparent increasing inflation, notwithstanding the increasing cost of line energy for flowers with large petals. Notably, the inset of Fig. 5

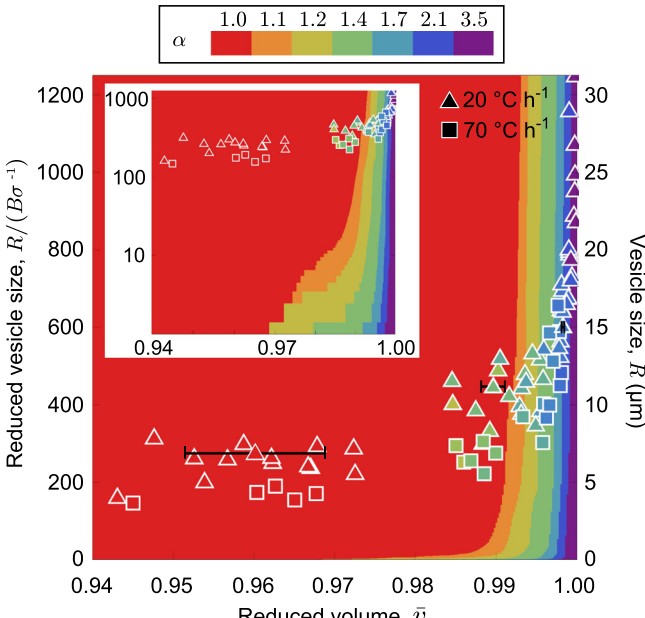

**Fig. 5 | State space with bending, line energy, and reduced volume.** Predicted state space of solid domain shapes (14% solid content) as a function of vesicle inflation and size. Petal formation (i.e., $\alpha \geq 1$) becomes favorable when the relaxation of bending energy exceeds the line-energy ($\sigma$) cost for increasing the solid perimeter. The left axis shows predictions as a function of vesicle radius scaled by $B/\sigma$, while the right axis shows the corresponding vesicle radius assuming an estimate $B/\sigma = 25$ nm. Experimental results, with measured size $\alpha$ and $\bar{v}$ superposed on the model predictions, for two cooling rates: slow 20 °C h$^{-1}$ (triangles), and fast 70 °C h$^{-1}$ (squares). Inset is the same map on a log-linear plot. Error bars on the example data reflect the resolution for 0.4 μm (2-3 pixels) on the calculation of $\bar{v}$, which is larger for the smaller vesicles that tend to have compact domains.

highlights that, with the exception of extremely small vesicles, the choice of the value of line tension has minimal impact on non-compact shapes (i.e., $\alpha \geq 1$). More specifically, while estimates of the line tension might vary considerably[45,46] Fig. 5 suggests that the boundary between convex and non-convex solid domain shapes is weakly-sensitive to the value of $\sigma$, having little impact even when it is varied by an order of magnitude.

## Size-dependent tension in thermally contracting vesicles

While competition between bending and line energy gives a preference for elaborate flower-shaped crystals on vesicles that are inflated and at elevated tensions, additional considerations explain why these elaborate solid domains grow selectively on large vesicles while compact domains grow on smaller vesicles. Figure 4c suggests that greater inflation and larger membrane tensions must be experienced by larger vesicles during nucleation and growth of solid domains. Indeed, larger inflations and tensions on larger vesicles are expected when thermal contractions and permeability are considered. Vesicles cooled from the one phase region to nucleate and grow crystal domains experience thermal contractions, with the coefficient of thermal expansion, $\kappa \equiv \frac{1}{A}(\partial A/\partial T)_{\tau}$[22], that tends to increase tension because the aqueous vesicle contents contract much less upon cooling than the predominantly hydrocarbon membrane[45,47]. However, tension squeezes water from the vesicle, which tends to reduce tension itself. This transport process, occurring more slowly for large vesicles having a smaller surface/volume ratio, is governed by permeability $\mathscr{P}$, defined: $\partial V/\partial t = - A(2\tau/R)(\mathscr{P}/\rho)$, with $\rho$ being the density of water. Note here that the pressure driving force for water transport is given by the Laplace term $2\tau/R$, providing the connection to membrane tension. Balancing the rate of tension increase due to thermal contractions with

the rate of tension relaxation caused by water permeation reveals, for a constant rate of cooling, a steady state tension that grows with the cooling rate and square of vesicle radius,

$$\tau_{ss} \approx \left(\frac{\kappa\rho}{4\mathscr{P}}\right)|dT/dt|R^2 \qquad (1)$$

Worth noting, that with annealing near 50 °C in the one-phase region, substantial cooling into the two-phase envelope preconditions tension before nucleation. This explains how the tendency for flower versus hexagon formation is predetermined before the crystals are visible, and why domain shape is preserved during growth as temperature decreases further. Indeed, growth occurs over less than ten additional degrees of cooling inside the two-phase envelope, per Fig. 1b.

For values of $\kappa = 0.005$ °C$^{-1}$[47] and $\mathscr{P} = 2.8 \times 10^{-16}$ s μm$^{-1}$[48], the steady-state tension can be as small as 0.5 mN m$^{-1}$ for 10 μm diameter vesicles and as large as 18 mN m$^{-1}$ for 60 μm diameter vesicles cooled at 0.3 °C min$^{-1}$, as done here. The latter exceeds the lysis tension, which falls in the range of 7–10 mN m$^{-1}$[48]. Also, the flower-shaped domains may be on vesicles experiencing lower tensions than required for stripe solid domain formation, a different polymorph[16,20]. Thus if large vesicles are not destroyed, they undergo burst-reseal processes that maintain their membrane tensions near or below lysis conditions[49]. Indeed, micropipette aspiration within 20 min after cooling and transferring vesicles to an open chamber (a process allowing some tension decay) reveals near zero tension in small vesicles for instance with diameters less than 20 μm, and substantial tension for larger vesicles, in Fig. 6.

Finally, we note that the growth of steady state tension as $|dT/dt|$ $R^2$ from the estimate in Eq. (1) suggests that the same morphology transitions—from compact to non-convex flower domains—can be achieved at a different size range simply through a change in cooling rate. Indeed, this shifting of the transition from hexagons to flowers was seen, in experiments, to occur on smaller vesicles when the cooling rate was increased. This behavior, accessing a broader range of combined vesicle sizes and tensions by using different cooling histories is included in Fig. 5. For example, comparing similarly sized vesicles (along a horizontal cut). Those vesicles cooled more quickly and reside to the right, with greater inflations and higher alpha values, indicative of more elaborate flowers.

## Outlook

The work illustrates how the morphologies of 2D crystals growing in an elastic 2D fluid are controlled by a competition between bending energy and line tension in a manner that inverts the expected crystalline domain morphology dependence on curvature radius understood to control crystallization on spherical templates. Additional properties of the 2D fluid, its coefficient of thermal expansion and permeability, allow for tension and therefore bending energy to be systematically controlled, leading to scalable manipulation of 2D crystal morphology in vesicles of different sizes. The current study suggests further scaling that could produce at least thousands of such 2D crystals of controlled morphology in individual batches, amenable to separation based on vesicle size, or density gradients. Thus, we demonstrate how 2D membrane fluids such as phospholipids can be implemented to controllably produce targeted 2D crystalline morphologies based on their physical properties and select these morphologies without changing chemical composition.

Notably, this study points out the critical effect of the flexibility of the "curved template" on which solid domains grow. Prior theories[2,3,5,7] and experiment[4] have considered solid domains forming on rigidly fixed spherical templates, which makes the incorporation of Gaussian curvature-induced strains unavoidable. While, according to Gauss's theorema egregium, the average Gaussian curvature of giant vesicles is

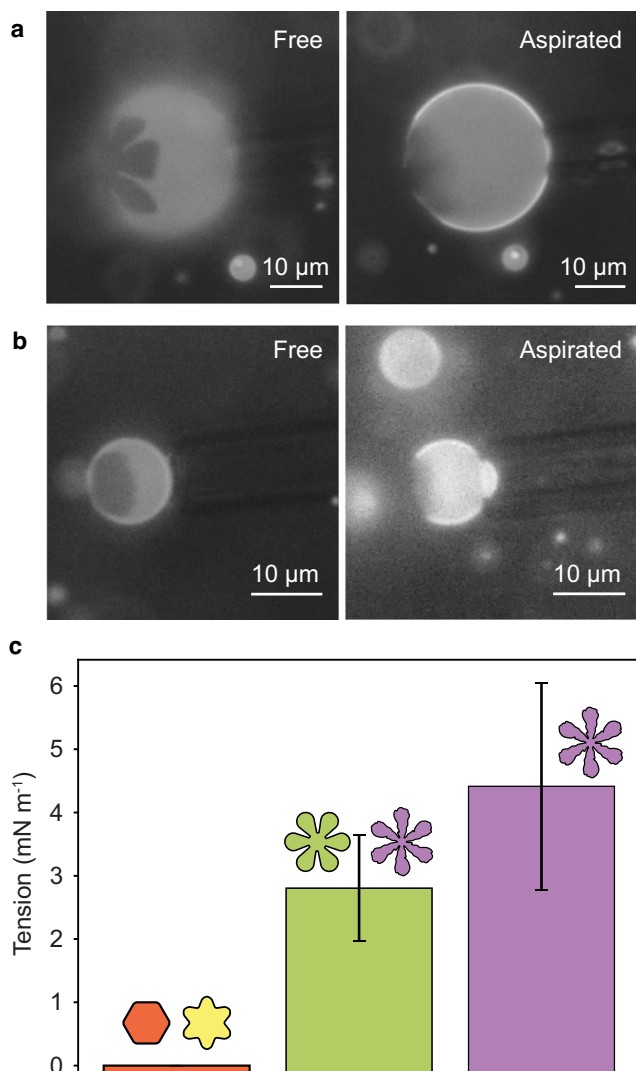

**Fig. 6 | Membrane tension assessment.** Micrographs of typical **a** flower-containing and **b** hexagon containing vesicles, before and after aspiration into a micropipette. Study was conducted within 20 min of cooling and crystal formation. The focal plane before aspiration shows the crystal pattern on each vesicle. After aspiration, the focal plane was adjusted to the equatorial plane of each vesicle to enable viewing of any membrane projection into the pipette. In **a**, the suction was relatively strong, 13.45 cm water (1.32 kPa), still not producing a projection and indicating high membrane tension. In **b**, a relatively low suction of 0.26 cm water (0.025 kPa) drew the membrane into the micropipette, indicating its negligible tension compared with the vesicle in **a**. **c** Tensions were measured within 20 min of cooling and domain formation. Samples from 8 different runs totaled sixteen vesicles with diameters under 20 μm, 11 vesicles with diameters in the range 20–40 μm, and two vesicles in excess of 40 μm. The latter were difficult to find. Color in each column represents the majority of crystal shapes present in the corresponding vesicle size range, including red (hexagon), green (simple flower), and purple (serrated flower). Error bars show standard error.

constant, the shape flexibility of vesicles allows for low energy bending modes to expel the topologically necessary Gaussian curvature from the solid into the fluid, giving rise to a distinct regime of shape-dependent elasticity energy of composite vesicles, with strain-free and nearly developable solid shapes.

The significance of these findings lies in their demonstration of a previously unknown mechanism to control the morphologies of

individual 2D crystals growing in elastic fluids. The vesicle-size-dependent growth and morphology of a rigid 2D solid were shown to be a result of the interplay between mechanics, membrane permeability, and thermal contraction, making a connection to membrane tension. Because tension comes about in biological systems through a variety of routes, there are potential implications for the organization of biological cell membranes. At the same time, the ability to grow select shapes of free-standing 2D crystals at scale may enable new routes for the processing of thin crystals for sensing, optical, and electronic applications.

## Methods

### GUV electroformation and phase separation

1,2-dioleoyl-sn-glycero-3-phosphocholine (DOPC, 99%), 1,2-dipalmitoyl-sn-glycero-3-phosphocholine (DPPC, 99%), and fluorescent tracer lipids, 1,2-dioleoyl-sn-glycero-3-phosphoethanolamine-N-(lissamine rhodamine B sulfonyl) (ammonium salt) (Rh-DOPE, 99%); and l,2-dipalmitoyl-sn-glycero-3-phosphoethanolamine-N-(lissamine rhodamine B sulfonyl) (ammonium salt) (Rh-DPPE, 99%) were purchased from Avanti Polar Lipids (Alabaster, AL). Dioctadecyl tetramethylindocarbocyanine perchlorate C18-DiI was purchased from Thermo Fisher Scientific. Vesicles having a 30/70 weight ratio (31/69 molar ratio) DPPC/DOPC plus 0.1 mol % Rh-DOPE, were electroformed on platinum (Pt) wires as previously detailed[16] but with the sucrose preheated and the electroforming temperature maintained in the range 55-70 °C to ensure vesicles were formed in the one phase region of the phase diagram, all having the same membrane composition. As a control, vesicles containing 20 wt% DPPC/80 wt% DOPC plus tracer were additionally studied to confirm an additional composition. Those images can be found in Supplementary Fig. 6. After electroformation, the stock vesicle suspension was harvested in a syringe and allowed to cool to room temperature for later use, within 2-3 days.

Studies of solid domain formation employed a 10-fold dilution of the stock vesicle solution in deionized (DI) water, which was transferred to a 10 mm × 10 mm closed chamber made from two coverslips and parafilm spacers. The chamber height varied in the range 0.1-0.5 mm depending on the desired solution volume (10–50 μL) for a specific experiment. The chamber was mounted on a custom-built temperature-control stage, heated to 55 °C for 5 min, cooled at 70 °C h⁻¹ to 42 °C, and then cooled at 20 or 70 °C h⁻¹ to room temperature, comprising the regular or fast cooling rates, respectively. A run with a single cooling ramp, at 0.3 °C min⁻¹ from 45 °C to room temperature in the Suppl. Fig. 7, demonstrates consistency with the two-step cooling in the main paper.

Vesicle and crystalline domain shapes were observed using a Nikon Eclipse TE 300 inverted epifluorescence microscope equipped with a 40× long working distance air fluorescence objective. Images were recorded with a pco.panda 4.2 sCMOS monochrome camera with a resolution of 0.17 μm per pixel at 40×, and analyzed using Nikon NIS Elements imaging software.

### Micropipette aspiration

Micropipette aspiration employed micropipettes pulled from glass capillaries to produce straight micropipette tips having inner diameters in the range 3-10 μm and flat ends. They were passivated with adsorbed bovine serum albumin before use. These were attached to a suction manometer equipped with a Validyne pressure transducer (model CD223) to control aspiration pressure and record values in the Nikon Elements Software. Micropipette experiments were conducted in an open-sided chamber consisting of two coverslips, passivated with adsorbed albumin, spaced by a microscope slide. After controlled cooling in the closed chamber to produce one solid crystal per vesicle, the liquid vesicle suspension was transferred to the open chamber[50] where it was held in place through capillary forces during micropipette studies which were conducted within 10-20 min after the end of the

cooling in the close chamber. The membrane tension was determined by employing the Laplace equation, as previously detailed[22].

### Thermodynamic modeling of solid-fluid composite vesicles

Our model considered vesicles of fixed total area $A$ and enclosed volume $V$, composed of area fraction $\phi$ of 2D solid and remaining $(1 - \phi)$ fluid membrane. The solid domain is modeled as an elastic plate with 2D Youngs modulus $Y$, Poisson ratio $\nu$, and plate bending modulus $B$,

$$E_{solid} = \frac{Y}{2(1+\nu)} \int_{solid} dA \left[ (\mathrm{Tr}\,\varepsilon)^2 + \frac{\nu}{1-\nu} \mathrm{Tr}\,\varepsilon^2 \right] + \frac{B}{2} \int_{solid} dA(2H)^2 \quad (2)$$

where $\varepsilon$ is the 2D strain and $H$ is the mean curvature. We consider the Helfrich model of the (fixed area) fluid region, with a bending energy $E_{fluid} = \frac{B}{2} \int_{fluid} (2H)^2$ where we assume the same bending modulus as the solid phase for simplicity. Because the 2D solid expels Gaussian curvature and the vesicle tangent remains continuous at the fluid-solid boundary, bending terms from the Helfrich energy that couple to Gaussian curvature in both fluid and solid phases are shape independent. That is, the area integral of the Gaussian curvature $K_G$ over the closed composite vesicle remains equal to $4\pi$ and, since the contribution from the solid domain is zero for any developable shapes, the $\int_{fluid} dA\, K_G = 4\pi$ for all configurations. While $K_G$ is not strictly constrained to be zero for the solid domains, Surface Evolver results shown in Fig. 4a confirm that in-plane solid elasticity leads to near-perfect expulsion of Gaussian curvature from the solid, which is consistent with the fact that over the relevant range of inflations, the solid strain energy is negligible (i.e., orders of magnitude below the bending as shown in Suppl. Fig. 8). On these grounds, the inclusion or absence of Gaussian bending modulus into the model has no discernable effect on the domain shape-dependence of elastic energy. Following simulation studies of Poisson effects in membranes[51], we chose $\nu = 0.4$ which falls within the range of predicted values. However, as thin membranes give rise to developable solid shapes (i.e., zero Gaussian) with essentially zero strain energy over the relevant range of experimental conditions (see Suppl. Fig. 8), the value of this parameter is expected to have little to no impact on results described here. Note that as a purely elastic (equilibrium) description of the system, the model energy does not include dissipative effects, such as membrane or solution viscosity.

We use Surface Evolver to minimize Eq. (2) subject to constraints of fixed solid and fluid phase area, and enclosed volume, and define reduced elastic energy $\bar{E} \equiv \frac{E_{solid} + E_{fluid}}{8\pi B}$ as the ratio of elastic energy of the composite relative to spherical fluid vesicle of the same bending stiffness.

We consider 6-fold symmetric solid domain shapes with variable petal-to-core aspect ratios. The shape of the solid domain is defined in a planar reference state. We model these shapes by the following family of radial functions:

$$r(\theta) = r_0 + \frac{1}{2} a r_0 \cos(6\theta) - \frac{1}{10} a r_0 \cos(12\theta) \quad (3)$$

where petal length is controlled through parameter $a$ (i.e., $\alpha = r(0)/r(\pi/6) = (5 + 2a)/(5 - 3a)$) and $r_0$ can be adjusted to fix the solid domain area. We describe the procedure for initializing closed meshes with solid domains of these shapes, subsequent energy minimization in Surface Evolver and characterization in the Suppl. Note 3.

For the effective core simulations, we hold a circular domain on the vesicle of a fixed area fraction $\Phi_{core}$ in a planar configuration. To determine the effective core for a given $\alpha$, $\bar{E}(\bar{v})$ is computed for the effective core model and $\Phi_{core}$ is adjusted so that minimal energy occurs for the same inflation $\bar{v}_0$ as the 14% solid with that $\alpha$ (see Supplementary Fig. 4).

The tension $\tau$ in the membrane can be related to the reduced energy $\bar{E}(\bar{v})$ dependence on inflation through the thermodynamic relation

$$\tau = -\left(\frac{\partial E}{\partial A}\right)_{V,\phi} = -8\pi B\left(\frac{\partial \bar{E}}{\partial \bar{v}}\right)_{\phi}\left(\frac{\partial \bar{v}}{\partial A}\right)_{V} = \frac{12\pi B}{A}\bar{v}\left(\frac{\partial \bar{E}}{\partial \bar{v}}\right)_{\phi} \quad (4)$$

To access the thermodynamic stability of flowered shapes, we consider the total energy

$$E = E_{\text{solid}} + E_{\text{fluid}} + \sigma P \quad (5)$$

where $\sigma$ and P are the line energy and perimeter of the boundary edge between the solid and fluid domains. The state map in Fig. 5 is determined by finding the value of $\alpha$ that minimizes the total energy in Eq. (5). When $t/R \rightarrow 0$ the elastic energy is strictly strain-free and derives from (mean-curvature) bending only. As in the case of fluid vesicles, this isometric limit is independent of vesicle size, depending only on $B$ and dimensionless ratios, $\bar{v}$, $\phi$, and $\alpha$. The line energy, however, is proportional to vesicle (and solid domain) size. Hence, as noted previously, comparing elastic and line energies requires comparing the vesicle radius to the length scale $B/\sigma$[45,46,52]. For mapping to the state space in Fig. 5, we employ an estimate for the bending modulus $B = 25$ $k_B T$ (corresponding to the fluid phase stiffness) and an estimate for line energy $\sigma = 1$ $k_B T$ nm$^{-1}$.

### Reporting summary
Further information on research design is available in the Nature Portfolio Reporting Summary linked to this article.

### Data availability
The data that support the findings of this study are available via the ScholarWorks at UMass Amherst[53] and from the corresponding author upon request.

### Code availability
The custom codes generated in this study are available via Scholar-Works at UMass Amherst[53] and from the corresponding author upon request.

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

## Acknowledgements
This work was supported by DOE DE-SC0017870.

## Author contributions
M.M.S. conceived and oversaw the experimental program and the thermal contraction /membrane permeation mechanism, performed calculations of tension evolution and steady state, developed data analysis strategies, and wrote most of the paper. G.M.G. conceived and oversaw the modeling of membrane mechanics, developed figures, and wrote part of the paper. H.W. executed the majority of experiments and data analysis, and developed the experimental Figures with W. X. conducting and analyzing micropipette experiments. G.J. executed calculations of membrane mechanics and developed Figs. based on modeling.

## Competing interests
The authors declare no competing interests.
