## [Peer Review File · Nature Communications]

Flower-shaped 2D crystals grown in curved fluid vesicle membranesEditorial Note: Parts of this Peer Review File have been redacted as indicated to maintain the confidentiality of unpublished data.

REVIEWER COMMENTS

Reviewer #1 (Remarks to the Author):

The manuscript contains a study on how the shape of phase-separated gel lipid domains on fluid unilamellar vesicles are affected by the vesicle size.

This is, in principle, good work given what is known on the subject; additional measurements (as described /suggested below) should be performed to make it publishable.

Most noteworthy results:

The main claim/focus of the study is that the size of vesicles 'dictates' the shape of the phase-separated domains: from circles (in smaller vesicles) to flower-like (in larger vesicles). Phase transitions/"crystallization", are driven, in this study, by the preferential saturated DPPC acyl-tail packing among themselves leading to the formation of these domains.

This correlation (vesicle size vs. domain shape) is, I believe, new in the field. The geometric characterization of phase-separated domains on fluid giant unilamellar vesicles with shapes ranging from circles to flower-like, have been reported before in a systematic study (Bandekar et al 2012, Langmuir), where the shape characterization metrics were also followed by the present manuscript (circular cap/core size beyond which scarring and floret shapes start to immerse etc). That study investigated different compositions and different types of phase-separating lipids, demonstrating some generality of the process where, beyond a domain surface area (size), for vesicles to exist, the ordered domain had to evolve in stripes of almost zero Gaussian curvature, with each stripe just following the bending (single curvature) of the vesicle (as also been predicted by Lipowsky, R.; Dimova, R. J. Phys. Condens. Matter 2003, 15, S31–S45.) Quoting from that article's abstract: "Domains phenomenologically comprise a circular "core" cap beyond which interfacial instabilities emerge resembling leaf-like stripes. At pH 5.0 stripes are of almost vanishing Gaussian curvature independent of GUVs' preparation path and in agreement with a general condensation mechanism. Increasing incompressibility of domains is strongly correlated with a larger number of thinner stripes per domain and increasing relative rigidity of domains with smaller core cap areas."

Significance in the relevant field: the findings would be more generalizable if more than a single composition of lipids (30:70 DPPC:DOPC) had been used in the study.

Support of conclusions: This reviewer disagrees with the hypothesis that the order of 2D 'crystals' formed are uniform across shapes and within each domain shape. Based on reported studies, this may well be an oversimplification: this is a strong assumption that in my opinion requires experimental evidence to support it.-- Given that the different domain shapes are a demonstration of different "mechanical" properties within the domain (bending/rigidity) - in particular: (1) the "round core"-like domain has been previously reported to be somewhat bended on both curvature directions (as also shown in Bandekar et al, Figure 1) ; whereas (2) the flower leaves have almost vanishing Gaussian curvature - then most probably the "crystals" cannot be assumed to be of same packing in all shapes and/or within the same shape, if the latter involves both a round core and leaves. In other words, these are most probably different crystal structures with possibly radial symmetry in packing order.

My suggestion is that the authors run additional supporting experiments to substantiate their claims/hypotheses - In particular,

(1) In addition to using a trace fluorescent lipid that only partitions in the fluid phase, all critical experiments should be repeated using a fluorescent lipid tracer that preferentially partitions in the ordered phase (that is called "solid" phase in the manuscript) - these images will most probably demonstrate the non-uniform microdistributions/intensities of the tracer within the domains (see Figure 1 vs Figure 2 in Bandekar et al 2012, Langmuir): a demonstration of non-uniform

'crystallization order" etc which can then be utilized both in the presented calculations and on Figure 2B –

(2) Page 9, lines: 163-164: "...the shape of the fluorescently labeled L_α fluid membrane suggests a primarily flat or cylindrically curved solid domain shape.." – as in (1), the authors should consider repeating the experiment including a tracer partitioning in the domains to clearly image the lower bending within the stripes of the flower-like domains, and not to need to make assumptions.

(3) Page 13, line 222 "...the fraction of Phi_{core} is modeled as rigid planar disk". Please see comments above. Also, if the rigid disk were entirely planar, then the scarring would be expected to start at radial distances closer to the center of the core (see Lipowsky, R.; Dimova, R. J. Phys. Condens. Matter 2003, 15, S31–S45.) – by changing the thickness of the 2D crystal, this matter could be easily resolved (please see below)

(4) The ratio of the two lipids should be varied to investigate the generality of the statements made: The domain-forming lipid, DPPC, is only 30% of the composition; at least one more composition with more DPPC should be used to investigate the domain shapes at conditions when there is, preferably, abundance of DPPC (the domain forming lipid)

(5) Alternatively to (4), the length of the saturated (domain forming lipid) could be varied to still allow the vesicles to be observed at room temperature but also work at conditions with excess domain forming lipid – at the same time, the 'thickness' of the 2D crystals will be changed (i.e. its bending rigidity and/or compressibility which will affect the core radius etc), better informing / understanding of the phenomenon.

(6) Page 3, line 65: "nearly pure ordered DPPC solid domains" – there is no experimental support for this statement; maybe consider replacing the word 'pure' with 'enriched' or something along these lines

(7) From the abstract: "...large vesicles sustain greater inflation and tension because their smaller area to volume ratio slows water permeation". If this is a kinetic constraint, then at longer times would this effect become attenuated/ vanish? Is it possible to observe the smaller vesicles at later times? – in the same context, maybe investigation of the effect of the reported limited expansivity of the fluid lipid membrane might play a critical role on the observations in figure 1C (i to iv), as reported by Devaux, P. F. Ann. Rev. Biophys. Biomol. Struct. 1992, 21 (1), 417–439. One suggestion would be to uniformly challenge the shapes/vesicles by introducing positive transbilayer osmotic gradients.

Minor suggestion

In some cases, the manuscript employs too convoluted a language that makes comprehension difficult. For example: "mitigation of solid strain drives formation of crystals with developable shapes that expel Gaussian curvature into the 2D fluid". Or in simple words: formation of domain shapes with vanishing Gaussian curvature.

The term 'crystal' may be attractive to general audience, but it is not clear if it would align with terms in the lipid biophysics community. Also the term 'solid' is used to indicate 'gel'. Maybe somewhere in the manuscript, the authors consider defining the precise meaning of these terms as they use them.

Reviewer #2 (Remarks to the Author):

The present manuscript by Wan et al deals with the formation of flower-shaped crystalline domains on unilamellar vesicle membranes. While such domain shapes have been reported in previous studies (Ref. 15), the mechanism of their formation remained unclear. The present studies propose such a

mechanism and test it by a comparison between a simple theoretical model and experimental observations.

The proposed mechanism attributes the observed flower shapes to geometrical frustration of the crystalline domains, which are incompatible with the Gaussian curvature. The fundamental and basic nature of the mechanism proposed by the authors for the present effect renders this work important for a wide scope of disciplines.

This mechanism's fundamentality serves as a captivating showcase of the geometrical principles that underlie complex phenomena in chemistry, biology, and beyond.

Furthermore, vesicles are widely regarded as a simple model of cell membranes, with the crystalline domains mimicking lipid rafts, potentially playing an important role in cell biology. Therefore, the elucidation of the mechanism which shapes these crystalline domains is of a significant importance for biology and possibly even for medicine.

In view of these considerations, I strongly recommend the manuscript to be accepted to Nature Communications, provided that the authors constructively address the major concerns which I have as for the validity of their conclusions. I do appreciate that adequately responding to my criticisms may require additional experimental work to be carried out. Unfortunately, the present version of the manuscript fails to convince me that the mechanism of flower-shaped domain formation is correct.

Major comments

=====

1. The most important distinguishing feature of the observed phenomenon is its anomalous dependence on the vesicle size (Line 50, Line 123-128): the high-R (low-curvature) vesicles exhibit flower-shaped domains, while the low-R (high-curvature) ones exhibit the most compact domain shapes.

The manuscript attributes this anomalous R-dependence to the membrane tension being much higher for the big vesicles. Therefore, the membrane tension is, arguably, the most important parameter in the experiment. Yet, this all-important parameter remains experimentally unknown. Instead of measuring this parameter, the manuscript presents a qualitative demonstration of it being higher for one specific big droplet, compared to one small droplet (Fig. 6). This single droplet experiment cannot replace a systematic measurement of this most important parameter, which could potentially allow the mechanism of domain shaping to be unequivocally determined.

While in some cases theoretical calculation of central parameters may be adequate, the dependence of membrane tension on R is quite complex: small tensions scale as R^2 (Eq. 1), while large tensions are probably R-independent, owing to water leakage at lysis-exceeding tensions. This complex dependence on R clearly calls for direct measurements of the tension. While the setup used by the authors apparently does not allow micropipette aspiration to be carried out in situ (line 297-298), the great importance of droplets' tension for the present studies may possibly justify setup modification.

2. If direct determination of tension is completely impossible, the proposed steady state dependence of the tension (Eq. 1) should at least, be tested by direct experiments, carried out for a range of dT/dt values (rather than just two examples).

Particularly informative would be the behavior at $dT/dt \rightarrow 0$ (quasistatic regime): does the R-dependence of domain shapes disappear upon the thermal equilibration of the sample?

While the fundamental theoretical model considers an equilibrium balance between bending and line tension, the experiments are only carried out at the steady state water leakage conditions. For the theoretical

mechanism

to be validated, it is crucial to repeat the experiments at (or near) the equilibrium conditions.

3. The experimental cooling protocol (lines 354-357) is too complex. It involves heating for a short time to a high temperature, then cooling at one rate, then cooling at a different rate. Since the rate of water squeezing out of thermally-compressed vesicles controls their tension and, consequently, shapes the crystalline domains, a simpler protocol should be adopted: pre-equilibration at a high temperature, followed by cooling at just one, well-defined, rate.

4. Figure 3 is supposed to demonstrate "a primarily flat or cylindrically curved solid domain shape" (lines 164-167). In addition, it should exhibit a developable folding, with the Gaussian curvature focused at generator intersections (lines 201-202). However, since the solid is not fluorescent and appears dark, one can only speculate about its exact shape and folding. A different method of visualization is needed here. Could bright field microscopy or a fluorescent staining of the aqueous phase help here?

5. The theoretical model does not allow the crystalline domain to adopt any other shape, beyond a 6-fold symmetric flower shape with a variable petal to core aspect ratio (Eq. 3). Thus, highly anisotropic protrusions and stripes, observed in some previous works (Refs. 5-8) are excluded. Topological structure defects are excluded as well. Thus, the fact that the model "reproduces" the experimental flower shapes does not prove the validity of the theoretical model: the shapes are simply built in. Since the validation of the domain shaping mechanism is the main goal of the present manuscript, and since the experimental R-dependence is complicated by the above-mentioned issues (due to the non-equilibrium aspects of the process), it could help a lot if the authors show that the fundamental equilibrium theoretical considerations do indeed lead to the formation of 6-fold symmetric flower shapes, without these shapes being built "by hand" into the model.

Minor issues and suggestions

=====

6. The manuscript should discuss the reason for the absence of flower shapes in other works, where crystals are similarly frustrated by the Gaussian curvature (e.g., Refs 5-8). This may have to do with the curvature being tunable for the presently-studied vesicles, while the curvature is fixed in other works, where e.g. colloidal particles reside on a droplet, the spherical shape of which is dictated by its relatively-high surface tension.

More surprising is the apparent absence of topological defects in the present system. The manuscript states that "the formation of topological defects that form in some curved 2D crystals[34, 35, 36] can only relax some, but not all, of the ... costs of Gaussian curvature." This statement is not completely correct. Note that the strain of the perfect icosahedron-

shape two-dimensional crystalline shell, with 12 disclinations residing at the vertices, is zero. Thus, one could imagine the growing developable crystalline domains in the presently-studied system to follow the shape of such a perfect icosahedral shell, incurring no strain-related penalties. This point should be discussed in the manuscript.

A more advanced theoretical model, called for in #5 (above), may allow the conditions for icosahedral, highly anisotropic stripe-like, and flower-shape domains to be delineated.

7. The theoretical alpha values, shown as a color map in Figure 5, exhibit no dependence on R when the reduced vesicle size is $\gg 1$. This is hardly surprising, as $R/(B \sigma^{-1})=1$ is the crossover between the bending- and the line tension- dominated regimes. I suggest that the authors consider changing the range of R-axis accordingly: one can either use a logarithmic R scale or introduce an axis break, allowing the high-R range to be compressed.

8. The relatively big gap in the experimental data in Figure 5 ($0.973 < \bar{v} < 0.984$) should be filled or explained, if possible. The gap is even larger for the squares. Is it a result of having the experimental data somehow thresholded?

9. The difference between the 20 degC/hour data and the 70 degC/hour data in Figure 5 is difficult to see. Consider adding a trend line or changing the R range to stronger emphasize the difference between these data sets.

10. Why is the Poisson ratio taken equal to 0.4? What happens for other values?

11. What breaks the 6-fold rotation symmetry of H and KG curvature distributions (Figure 4A) for $\alpha > 1.1$? The solid is 6-fold symmetric, so that it is surprising that H and KG are not symmetric under horizontal rotation about the center of the solid domain. Remarkably, the $\alpha=1$ distributions, where the solid is circular, seem to exhibit a 4-fold symmetric H and KG distributions. The origin of these symmetry breaking effects should be discussed in the manuscript.

12. The generators in Figure S9 (dashed lines) end, suspiciously, precisely at the sawtooth-like discontinuities which originate in the finite mesh size of Surface Evolver simulations. I wonder, whether these generators stay at the same positions or even survive, if a finer mesh is adopted for the simulations.

13. The cartoon in Fig. 1A is not clear. The cut in the bilayer has a different orientation in the main panel and in the inset. The 3D perspective is incorrect, so that the cartoon's meaning is difficult to understand.

14. It is not clear, what are: "slightly inflated conditions (i.e. \bar{v} above the minimal elastic energy state)" (Line 186)? The value of \bar{v} used in the calculation should be specified.

15. The calculation of the tie line should either be completely removed to the Supplemental

Information, together with Figure 1B (!), or explained more clearly in the main text. The present explanation is only good for the experts, not for the general Nat. Commun. readers.

16. I recommend the authors to consider completely removing the discussion of the effective core model to the Supplemental Information. The universal behaviour of $I(\tau)$, while interesting for itself, contributes very little to the main topic of the paper.

17. Why is the $E(\text{strain})$, in Line 150, proportional to ϕ^3 ? The dependence on ϕ does not seem to be important in what follows. Therefore, I recommend the authors to choose whether to explain it properly or remove it (keeping just the R -dependence).

18. I recommend that Line 182, explaining the dimensionless measure of inflation, comes before the words "under nearly inextensible conditions...", to help the reader.

19. Line 180: "with a minimum at a particular inflation ... that decreases over a series of solid shapes with increasingly larger petals (increasing α)." This statement should be clarified. The minimum gets deeper with α . The v_0 value INCREASES with α . It is not clear, what quantity does "decrease" for increasing α .

20. Line 32 in the Supplemental Information: using R for two different quantities is somewhat misleading.

21. I also spotted several typos:

Line 213: "corresponding tensed"

Line 241: "conform to more uniformly to"

Line 398: "defined it perimeter"

Line 402: "adjusted to fixed the solid"

Line 416: "where σ " and P "are the line energy and perimeter"

The manuscript by Wan et al., on "Flowering of Developable 2D Crystal Shapes in Closed, Fluid Membranes" presents valuable insights into how the morphologies of 2D crystals growing in an elastic 2D fluid are controlled by a fundamentally new mechanism. There are some exciting new observations: the authors demonstrate a strong correlation between domain shape and vesicle size, while they use the same unique chemical composition of vesicles throughout the experiments. Furthermore, they present a detailed energetic model that elucidates the vesicle size-based selectivity of crystal morphology. The manuscript is well written, and the authors have conducted commendable work which may have a significant importance for related fields in science.

Discussed below are several things that require further attention to improve this version of manuscript:

Methodology and Data:

The study design and data analysis seem robust and appropriate for the research questions addressed. I'd like to suggest testing another set of chemical composition of the fluid membrane to

ensure the claimed conclusion (i.e., vesicle size selects crystal shape). Although the data contains over 330 vesicles (manuscript line 117), validating the mechanism with another lipid composition (either a different bulk concentration of DPPC or a different pair of lipids) would strengthen the validity of the findings. If there is a particular reason for the specific DPPC concentration (30%), this point should be clearly explained in the text.

Significance:

The paper addresses an important issue in the field of curved two-dimensional crystals, but it has a lack in explaining the potential uses and applications of the phenomenon under investigation. Providing a comprehensive discussion on the potential real-world implications and practical applications of the discovered phenomenon would significantly enhance the manuscript's value to the Nature Communications' readers and further emphasize its significance for the broader scientific community.

In addition, I strongly recommend inclusion of a short paragraph that elaborates on the importance of this phenomenon for other related fields of science. This could provide valuable insights and further enrich the discussion of tangent fields in the manuscript.

Simulations:

Based on my understanding of the theoretical model, it appears that the presence of the 6-fold symmetric flower domain shape is predetermined and fixed within the model. Unfortunately, the model does not consider the influence of topological defects or allow the domain to adopt a preferred shape based solely on energetic considerations. As a result, the model is unable to provide the sufficient support for the discussed approach mechanism.

Additional comments:

The figures and graphs included are informative and have detailed captions, but I suggest a slight change in Fig 5 since it's hard to tell the difference between the two cooling rates: triangles and squares. Trend lines might help.

Overall, the manuscript is organized logically and coherently, and I find it interesting and valuable. I strongly recommend it to be accepted, given that the authors will take my comments into consideration.

Reviewer #3 (Remarks to the Author):

I co-reviewed this manuscript with one of the reviewers who provided the listed reports as part of the Nature Communications initiative to facilitate training in peer review and appropriate recognition for co-reviewers.

Reviewer #4 (Remarks to the Author):

The manuscript reports the results of a detailed experimental study of the morphologies of two-dimensional (2D) crystals, nucleated, grown, and integrated within the fluid membrane of giant vesicle. The shape of the crystal domains is found to be dictated by the interplay of mechanics, membrane permeability, and thermal vesicle contraction.

The cost of solid strain of crystalline domains with Gaussian curvature drives formation of crystals with locally planar or cylindrical shapes, which expels Gaussian curvature into fluid membrane regions. Upon cooling to grow the crystals, large vesicles are found to sustain greater inflation and tension, which leads to the formation of more elaborate shapes on large vesicles despite their more gradual

curvature, while small vesicles display compact planar crystals. This size dependence is opposite to the behavior of crystals on rigid spherical surfaces.

The interplay of curvature elasticity of membranes, the in-plane stretching elasticity of crystalline, the domain formation in multicomponent systems, and the membrane deformation in three dimensions gives rise to a multitude of interesting and often unexpected properties of vesicles. In the current manuscript, it is the expulsion of Gaussian curvature from crystalline domains coupled with the induced curvature of the embedding fluid membrane which generates intriguing flower-like domain shapes, which differ strongly from those on spherical geometries. Thus, the current study significantly enhances the understanding of complex membrane behaviors.

The following questions and comments should be addressed:

(1) The authors emphasize that "... morphologies of 2D crystals growing within an elastic 2D fluid having a flexible shape and a closed topology ... are controlled by a fundamentally distinct mechanism (than for fixed spherical templates)".

Similarly, they state:

"Remarkably, the most compact shapes are found on the small vesicles, i.e. those with the greatest curvature, while non-convex flowers are seen on larger vesicles, having smaller curvature. This correlation runs counter to that show the threshold size for the transition from compact to anisotropic domain shape increases with larger sphere radius".

I think that in this form and generality, these statements are somewhat misleading. The reason is that IN THE DESCRIBED EXPERIMENTS, it is not only the membrane curvature and deformability which determines the observed shapes, but equally important the thermal membrane contractions and water permeation, which generates a size-dependent membrane tension. This should be discussed more transparently.

(2) What is the role of line tension? In the current system, the line tension is clearly very small. For larger line tension, compact domains or even domain budding should be expected, see, e.g. R. Lipowsky, J. Physique II, 2, 1825 (1992); R.J. Mashl et al., Biophys. J. 74, 2862 (1998); T. Kohyama et al., Phys. Rev. E 68, 061905 (2003).

(3) In the spherical limit (of reduced volume approaching unity, or large tension), previous results for spherical surfaces should be recovered. From the results for spherical surfaces, I would expect petals to become stripe solid domains. Thus, it seems to me that petals and stripe solid domains are actually closely related polymorphs.

(4) The effective core model seems to imply that petals deformation energy becomes negligible in the spherical limit (high tension). Doesn't this again indicate the close relation of petals and stripe solid domains on spherical geometries?

(5) Two related phenomena, which the authors might want to briefly discuss, are:

-- Grain boundary scars in curved surfaces with crystalline order:

** rigid spherical shape: A.R. Bausch et al., Science 299, 1716 (2003);

** deformable vesicles: T. Kohyama et al., Phys. Rev. Lett. 98, 198101 (2007).

-- Multicomponent crystalline vesicles with different mechanical properties of the components:

** M.D. Emanuel et al., Phys. Rev. E 102, 062104 (2020);

** S. Li et al., ACS Nano 15, 14804 (2021).

Reviewer #5 (Remarks to the Author):

The manuscript considers growing crystals on flexible membranes and claims possibilities to control size and shape by the size of the membrane and growth conditions. The observed phenomena are nicely explained by reviewing theoretical results ranging from crystal growth on spherical surfaces, to elastic bending and stretching theory and their relation to intrinsic and extrinsic curvature. The experimental setup somehow can be seen as a realization of these theories and in this clearness I'm not aware of any other such study. I only have some concerns with the considered simulations. They are based on Surface Evolver and the considered setting is probably the best one can do within this software. However, the authors should at least mention and discuss the limitations.

1) The considered 6-fold symmetry is an input, the model does not contain such information. In [6,7] this symmetry results from the microscopic lattice structure.

In case there is a possibility to measure this experimentally, it would be nice to make a statement on possible defects in the solid phase, or stress distributions of these solids to confirm the relation to the theories [6,7].

2) The relaxation, seems not to consider any relaxation in the phases, e.g. there is no line tension considered. While it is argued that the magnitude is much lower, a simultaneous relaxation of the phases and the shape leads to various additional coupling terms, which should not be ignored, see e.g. Bachini et al. arXiv:2305.15147 for a recent discussion in the context of two fluid phases.

3) It should be explicitly stated that the simulations do not consider growth or any dynamic evolution.

4) It is argued that the bending rigidity is set to be the same for solid and fluid phase for simplicity and the flattening of the solid phase is associated with the considered plate theory. However, also different bending energies already could lead to this effect, see again Bachini et al. arXiv:2305.15147. The plate theory would not be needed in this case. This should be discussed. I guess it could also be simulated within the Surface Evolver.

5) [4,7] should not be called phase-field modeling, but phase-field-crystal modeling, as the microscopic origin is essential.

Response to Reviewers

We thank the reviewers for their close read of our manuscript and comments that will improve this paper. We have conducted extensive additional work to address their comments. Some of the new work appears specifically in the paper (Figure 3 has been replaced with images from new experiments. Figures 4 and 5 have been modified and Figure 6 includes new data on membrane tension). Also in response to a reviewer suggestion, Figure 1 has been updated.

In other instances, results from new experiments appear in the Supporting Information. This category includes new studies with two additional tracer dyes, required by reviewers, which uphold and strengthen our original conclusions.

In other cases, additional experiments were conducted to address the concern of a reviewer concern but the findings themselves were off topic from off-point of the paper itself. Those results are included here in the response to reviewers, not for publication. These include studies at an alternate cooling history and images of vesicles containing a greater proportion of DPPC.

Figure RR-1. Distribution of crystal shapes on vesicles of different sizes for a one step cooling history of 0.3 C/min. The one step cooling study, including 151 vesicles in a single run, was conducted in response to reviewer 2. The two step cooling data include 330 vesicles from 3 runs.

Reviewer #1 (Remarks to the Author):

The manuscript contains a study on how the shape of phase-separated gel lipid domains on fluid unilamellar vesicles are affected by the vesicle size.

This is, in principle, good work given what is known on the subject; additional measurements (as described /suggested below) should be performed to make it publishable.

Most noteworthy results:

The main claim/focus of the study is that the size of vesicles ‘dictates’ the shape of the phase-separated domains: from circles (in smaller vesicles) to flower-like (in larger vesicles). Phase transitions/”crystallization”, are driven, in this study, by the preferential saturated DPPC acyl-tail packing among themselves leading to the formation of these domains.

This correlation (vesicle size vs. domain shape) is, I believe, new in the field. The geometric characterization of phase-separated domains on fluid giant unilamellar vesicles with shapes ranging for circles to flower-like, have been reported before in a systematic study (Bandeekar et al 2012, Langmuir), where the shape characterization metrics were also followed by the present manuscript (circular cap/core size beyond which scarring and floret shapes start to immerse etc). That study investigated different compositions and different types of phase-separating lipids, demonstrating some generality of the process where, beyond a domain surface area (size), for vesicles to exist, the ordered domain had to evolve in stripes of almost zero Gaussian curvature, with each stripe just following the bending (single curvature) of the vesicle (as also been predicted by Lipowsky, R.; Dimova, R. J. Phys. Condens. Matter 2003, 15, S31–S45.) Quoting from that article’s abstract: “Domains phenomenologically comprise a circular “core” cap beyond which interfacial instabilities emerge resembling leaf-like stripes. At pH 5.0 stripes are of almost vanishing Gaussian curvature independent of GUVs’ preparation path and in agreement with a general condensation mechanism. Increasing incompressibility of domains is strongly correlated with a larger number of thinner stripes per domain and increasing relative rigidity of domains with smaller core cap areas.”

Significance in the relevant field: the findings would be more generalizable if more than a single composition of lipids (30:70 DPPC:DOPC) had been used in the study.

Support of conclusions: This reviewer disagrees with the hypothesis that the order of 2D ‘crystals’ formed are uniform across shapes and within each domain shape. Based on reported studies, this may well be an oversimplification: this is a strong assumption that in my opinion requires experimental evidence to support it.-- Given that the different domain shapes are a demonstration of different “mechanical” properties within the domain (bending/rigidity) - in particular: (1) the “ round core”-like domain has been previously reported to be somewhat bended on both curvature directions (as also shown in Bandeekar et al, Figure 1) ; whereas (2) the flower leaves have almost

vanishing Gaussian curvature - then most probably the “crystals” cannot be assumed to be of same packing in all shapes and/or within the same shape, if the latter involves both a round core and leaves. In other words, these are most probably different crystal structures with possibly radial symmetry in packing order.

Au response: Thanks for such comprehensive input. We wish to make a few general points in response.

First, we intend our work to demonstrate key features of 2D crystallization in 2D elastic fluids, and employ giant unilamellar vesicles as a platform. A key takeaway is not only the size dependence of the crystal morphology, but the fact that it opposes crystallization on templates of non-zero gaussian curvature that may lack the bending energy and tension at the crystal edge, inverting what was found in these other crystallizing systems and modeling efforts.

Next, we appreciate that our system and results support the idea of a uniform mechanical properties of the solid crystal, but we do not make such a hypothesis or conclusion in the manuscript, misstated by the reviewer. The solid phase of the surface evolver calculations is indeed uniform, and elasticity does not vary across the solid. The results of those calculations are shown and compared, through the use of a state space and qualitative vesicle shapes to the experiments. Our experimental design, results, or conclusions do not make the hypothesis stated by the reviewer. In fact, by modeling the base case that lacks any variation of mechanics in the solid phase, we reveal when compact versus flower domains are preferred in the base case situation. Any complexities (anisotropy, spatial variations) in the solid domain mechanics would build on this, should they be interesting to add in future works, when needed. Here the simple model, not requiring substantial variation in solid properties, captures the essential features of the experiments.

We have added the Lipowski/Dimova 2003 in the introduction.

Finally, we have done detailed additional experiments to address this reviewers concerns, especially in Figure 3.

--

My suggestion is that the authors run additional supporting experiments to substantiate their claims/hypotheses - In particular,

(1) In addition to using a trace fluorescent lipid that only partitions in the fluid phase, all critical experiments should be repeated using a fluorescent lipid tracer that preferentially partitions in the ordered phase (that is called “solid” phase in the manuscript) – these images will most probably demonstrate the non-uniform microdistributions/intensities of the tracer within the domains (see Figure 1 vs Figure 2 in Bandekar et al 2012, Langmuir): a demonstration of non-uniform ‘crystallization order’ etc which can then be utilized both in the presented calculations and on Figure 2B –

Au response: We conducted the suggested experiments employing two dyes, in Figure SI-9. We find no incorporation of either new tracer into the solid phase in our study, which differs from

that in Bandekar et al. These new experiments show that crystallizing DPPC uniformly excludes both Rh-DPPE and C18-DiI, the latter employed by Bandekar. In summary the solid domains in this study exclude three tracers, Rh-DPPE, Rh-DOPE, and C18-DiI. This is a property of our solid and it further establishes that our solid is not the classic Lb' gel phase which is known to incorporate Rh-DPPE.^{1,2} We further note that incorporation of a tracer into a crystalline phase might disrupt solid structure and alter mechanics, creating the variation in mechanics which might be very domain specific and complicating. Some solid lipid phases, such as the Pb' phase are understood to exclude other lipids. While exclusion of a tracer eliminates an opportunity to image the solid directly, tracer exclusion is consistent with (but in no way proving) a uniform crystalline solid. We continue not to claim such uniformity crystal formation.

(2) Page 9, lines: 163-164: “..the shape of the fluorescently labeled L_alpha fluid membrane suggests a primarily flat or cylindrically curved solid domain shape..” – as in (1), the authors should consider repeating the experiment including a tracer partitioning in the domains to clearly image the lower bending within the stripes of the flower-like domains, and not to need to make assumptions.

Au response: We describe above how two tracers, the best likely candidates for incorporating into the solid failed to do so. In prior work^{1,3} we incorporated Rh-DPPE into striped DPPC Lb' domains, demonstrating that we are familiar with incorporation of tracer into solid domains when the solid polymorph is a type that accommodates the tracer. Indeed not all solids incorporate tracers, including the domains in this work, which are like the compact domains in our previous studies, which also excluded tracers.

We agree with the need to better image the solid regions of the vesicle to better establish the contour of the solid domains. To this end we have conducted additional studies employing phase contrast and bright field to image the vesicle contour, seen in a NEW **Figure 3**. These new results demonstrate the relative flatness or only cylindrical bending of the petals, and of compact domains. We do not see, within the limits of the imaging, evidence for cap shapes or other non-zero Gaussian curvature.

(3) Page 13, line 222 “...the fraction of Phi_core is modeled as rigid planar disk”. Please see comments above. Also, if the rigid disk were entirely planar, then the scarring would be expected to start at radial distances closer to the center of the core (see Lipowsky, R.; Dimova, R. J. Phys. Condens. Matter 2003, 15, S31–S45.) – by changing the thickness of the 2D crystal, this matter could be easily resolved (please see below).

Au response: We believe the reviewer is confusing the “effective core model” with the generic (flexible solid domain) model. The general model is actually allows with the solid model to bend and stretch with variable solid domain shapes (i.e. petal to core length ratios). These are the results shown in Fig. 4A-B. While we don't explicitly prohibit Gaussian curvature in the solid, as a thin solid, the relative expensive elastic energy of shearing required by Gaussian curvature evidently favors expelling Gaussian curvature from the solid to fluid phase in the vesicle. Hence, we don't impose the curvature in the general model to do anything, nevertheless, we find that the bending energy tends to organize spatially in a pattern that can be well

approximated by an “effective core” model, where the non-monotonic elastic energy dependence on inflation (\bar{v}) in Fig. 4B can be captured by imposing a planar circular core of suitable radius on an axisymmetric vesicle (inset of Fig. 4C). Our motivation to explore this simplified geometrical ansatz is two-fold. First, we want to demonstrate that the effect of petal formation is to allow for solid bending that more closely approximates the preferred spherical shape of the inflated, predominately fluid vesicle, and leads to a concentration of bending energy at the edge of the core region in tensed vesicle. Second, we exploited the effective core model to demonstrate a generic connection between increased inflation and tension in the vesicle, results which derive purely from the bending elastic costs. This latter point is essential for establishing the connection between selected solid shape (which depends on inflation) and vesicle size (which effectively controls tension).

(4) The ratio of the two lipids should be varied to investigate the generality of the statements made: The domain-forming lipid, DPPC, is only 30% of the composition; at least one more composition with more DPPC should be used to investigate the domain shapes at conditions when there is, preferably, abundance of DPPC (the domain forming lipid)

Au response: This paper deals with the nucleation and growth of 2D crystals in elastic 2D fluid and focuses on conditions where each crystal can grow without interference from other objects. To see these beautiful crystals, we invested effort to identify conditions (compositions, cooling rates) producing one crystal per vesicle, enabling clear results. (For instance, one crystal per vesicle enables some of the morphologies where the petals wrap substantially around the vesicle.) This point is now amplified on page 4 of the revised manuscript.

We very much appreciate the importance of showing a generalizable result. We therefore have conducted additional studies, included in the front of this reviewer response document and not for publication, for 60 mol% DPPC / 40 mol% DOPC. Greater DPPC content produces a greater solid area fraction on each vesicle, along with multiple nucleation sites, and hence domains, per vesicle.

Indeed, we might have expected this since it is established that, during cooling the number of nuclei and, ultimately, number of solid domains increases substantially with the DPPC/DOPC ratio.⁴ Multiple domains, which occur at greater DPPC concentrations, complicate the picture through several routes. First with multiple crystals, each crystal is small and grows slightly less extensively with cooling than what is observed with a single larger crystal, though small domains can still form flowers. The conditions are shifted considerably though. Also complicating, multiple crystals diffusing on a single vesicle, tend to approach each other and aggregate at high tensions. Closely approach domains apparently influence the shape of crystal growth, even when not touching, a topic for future work. Also important, with more DPPC in the membrane, at a given temperature in the two phase region the solid area fraction at a given temperature is greater. Besides producing more interference between neighboring crystals, the higher DPPC content and produces more membrane contraction (loss of fluid and creation of more area-dense solid), drives tension up, favoring more flowers. (The contraction due to the phase change is small in the current work.) Yet, crowding alters crystal shape and prohibits long petals which just

cannot fit on the vesicle. All these complexities, arising from additional physical mechanisms make it more difficult to isolate the behavior we report in the single domain case. That does not negate the mechanism we report.

(5) Alternatively to (4), the length of the saturated (domain forming lipid) could be varied to still allow the vesicles to be observed at room temperature but also work at conditions with excess domain forming lipid – at the same time, the ‘thickness’ of the 2D crystals will be changed (i.e. its bending rigidity and/or compressibility which will affect the core radius etc), better informing / understanding of the phenomenon.

Au response: It is not trivial to change lipid systems, altering the phase transition temperature and extent of cooling (and therefore membrane contraction) from nucleation down to room temperature. The suggestion to study different multiple lipid systems does not add to the understanding of new mechanism in this work, which targets a general mechanism and not just lipids. The current use of continuum theory rather than an experimental survey of different materials, we feel, is the more powerful way to demonstrate mechanism across of range of systems which could go beyond lipids. Further important to note our modeling addresses the simplest base case in order to establish the key features behind our observations: the difference between the fluid and solid phase lies is the fluid’s ability to shear and the solid’s rigidity against shear. The behavior does not result from a stiffer (less bendy) solid domains relative to the fluid.

(6) Page 3, line 65: “nearly pure ordered DPPC solid domains” – there is no experimental support for this statement; maybe consider replacing the word ‘pure’ with ‘enriched’ or something along these lines

Au response: It is well established in prior work that the solid is not simply enriched in DPPC relative to the fluid (as is seen in 3 component systems that make Lo and Ld phases), but that it is nearly pure. This is further supported by the inability of the current solid to incorporate a variety of tracers, and by re-measuring of the phase diagram. References have been added.

(7) From the abstract: “..large vesicles sustain greater inflation and tension because their smaller area to volume ratio slows water permeation”. If this is a kinetic constraint, then at longer times would this effect become attenuated/ vanish? Is it possible to observe the smaller vesicles at later times? – in the same context, maybe investigation of the effect of the reported limited expansivity of the fluid lipid membrane might play a critical role on the observations in figure 1C (i to iv), as reported by Devaux, P. F. Ann. Rev. Biophys. Biomol. Struct. 1992, 21 (1), 417–439. One suggestion would be to uniformly challenge the shapes/vesicles by introducing positive transbilayer osmotic gradients.

Au response: Yes, kinetics are important, as the reviewer suggests, and indeed, Figure 5 of the original ms included results at two different cooling rates which conform to a single state space,

but occupy slightly different regions as a result of the different inflation values that come about for different cooling rates. The reviewer asks about long times and yes, at long times the tension will vanish. However, the domains nucleate and grow during the cooling process and so long time relaxations at room temperature, after cooling and crystallization are complete, do not contribute to domain shape. Growth only happens as one moves down through the two phase region of the phase diagram. And yes, in principle it would be possible to favor the formation of compact crystals instead of flower shapes by using osmolyte to make the membranes very floppy so that tension never develops. However, doing this well is tricky because tethers can form that interfere with the more tidy behaviors we see here.

Minor suggestion

In some cases, the manuscript employs too convoluted a language that makes comprehension difficult. For example: “mitigation of solid strain drives formation of crystals with developable shapes that expel Gaussian curvature into the 2D fluid”. Or in simple words: formation of domain shapes with vanishing Gaussian curvature.

The term ‘crystal’ may be attractive to general audience, but it is not clear if it would align with terms in the lipid biophysics community. Also the term ‘solid’ is used to indicate ‘gel’. Maybe somewhere in the manuscript, the authors consider defining the precise meaning of these terms as they use them.

Au response: We have reviewed the manuscript for opportunities to simplify language without losing meaning for the different scientific communities with potential interest in this work.

We appreciate the general point of the reviewer with regard to the sentence in the abstract, although we have to emphasize that there are multiple research communities that we target with this article and to some extent challenges to translate across gaps are to be expected. For the sentence above, it is important to emphasize that not only does the 2D solid elasticity favor zero Gaussian curvature, but it “redistributes its Gaussian curvature” to the surrounding fluid phase, which in effect increases the bending energy of the composite vesicle. That is, it is important to point out that the average Gaussian curvature is constant due to a topological constraint, but it can be distributed unevenly into different phases depending on the mechanical cost. We believe the reviewer may be missing the importance of this second part of the mechanism and argue that “expelling” Gaussian curvature from the solid to the fluid is indeed the simplest and most concise way to encapsulate the mechanism. This is the essence of what we argue to be the novel mechanism at the heart of the size (tension) dependent morphology selection. To make this more clear, we have revised this abstract sentence as well as a similar sentence in the introductions.

Of note, specific to the lipid community, “solid” is not meant to indicate “gel” (the Lb’) phase. We know the solid is not the Lb’ gel because the current domains all exclude Rh-DPPE tracer which is incorporated into the gel Lb’ phase in other works. In the MS we do not discuss which polymorph we have because such discussion detracts from the scientific point and we wish to avoid specific assumptions about the nature of the solid polymorph. By using “solid” we connect to the mechanical picture, and specific definitions with regard to shear rigidity.

Additionally, as emphasized in response to several reviewer questions, the emergent 6-fold (e.g. hexagonal faceting) character of some of the domains suggests a further role of the underlying crystalline symmetry of the molecular ordering in solid domains.

Revised abstract sentence:

“Mitigation of solid strain drives formation of crystals with vanishing Gaussian curvature (i.e. developable) domain shapes and enhanced Gaussian curvature the surrounding 2D fluid.”

Revised sentence in 2nd paragraph of intro:

“The in-plane solid elasticity of 2D crystals, i.e. a preference for flatness or cylindrical bending,⁵
⁶ favors solid domains with zero Gaussian curvature. Because topology requires that the total Gaussian curvature of the composite vesicle is constant, the expulsion of Gaussian curvature from the solid domain must be redistributed to the 2D fluid,⁷ leading to a complex interplay between the shape and morphology of the crystal and the system’s global shape and bending energy.”

Reviewer #2 (Remarks to the Author):

The present manuscript by Wan et al deals with the formation of flower-shaped crystalline domains on unilamellar vesicle membranes. While such domain shapes have been reported in previous studies (Ref. 15), the mechanism of their formation remained unclear. The present studies propose such a mechanism and test it by a comparison between a simple theoretical model and experimental observations.

The proposed mechanism attributes the observed flower shapes to geometrical frustration of the crystalline domains, which are incompatible with the Gaussian curvature. The fundamental and basic nature of the mechanism proposed by the authors for the present effect renders this work important for a wide scope of disciplines. This mechanism's fundamentality serves as a captivating showcase of the geometrical principles that underlie complex phenomena in chemistry, biology, and beyond.

Furthermore, vesicles are widely regarded as a simple model of cell membranes, with the crystalline domains mimicking lipid rafts, potentially playing an important role in cell biology. Therefore, the elucidation of the mechanism which shapes these crystalline domains is of a significant importance for biology and possibly even for medicine.

In view of these considerations, I strongly recommend the manuscript to be accepted to Nature Communications, provided that the authors constructively address the major concerns which I have as for the validity of their conclusions. I do appreciate that adequately responding to my criticisms may require additional experimental work to be carried out. Unfortunately, the present version of the manuscript fails to convince me that the mechanism of flower-shaped domain formation is correct.

Major comments

=====

1. The most important distinguishing feature of the observed phenomenon is its anomalous dependence on the vesicle size (Line 50, Line 123-128): the high-R (low-curvature) vesicles exhibit flower-shaped domains, while the low-R (high-curvature) ones exhibit the most compact domain shapes.

The manuscript attributes this anomalous R-dependence to the membrane tension being much higher for the big vesicles. Therefore, the membrane tension is, arguably, the most important parameter in the experiment. Yet, this all-important parameter remains experimentally unknown. Instead of measuring this parameter, the manuscript presents a qualitative demonstration of it being higher for one specific big droplet, compared to one small droplet (Fig. 6). This single droplet experiment cannot replace a systematic measurement of this most important parameter, which could potentially allow the mechanism of domain shaping to be unequivocally determined.

While in some cases theoretical calculation of central parameters may be adequate, the dependence of membrane tension on R is quite complex: small tensions scale as R^2

(Eq. 1), while large tensions are probably R-independent, owing to water leakage at lysis-exceeding tensions. This complex dependence on R clearly calls for direct measurements of the tension. While the setup used by the authors apparently does not allow micropipette aspiration to be carried out in situ (line 297-298), the great importance of droplets' tension for the present studies may possibly justify setup modification.

Au response: The measurement of tension is generally difficult and more specifically impossible during cooling and crystallization: Tension changes with time and the temperature control chamber for a small fluid volume cannot accommodate the introduction of micropipettes, which introduce a large thermal mass, completely disrupt the (critical) cooling history, enable evaporation, etc. No experimental re-vamp will fix these problems (else we would have done them already.) After vesicles reach room temperature with their fully established crystal patterns, the tension decays in time as water is forced from high-tensioned vesicles. The decay is expected to occur more quickly for small vesicles. As the reviewer points out, when the tension is high, vesicles can burst, and undergo burst-reseal processes in time causing tension fluctuations in the most tightly inflated vesicles. We thought, therefore, that the images of a tight vesicle versus one that is looser one in the original ms would make the point visually. We appreciate that these are just two vesicles.

The best we can do for quantitation of studies are experiments that transfer the drop of liquid from the small chamber (needed for temperature control) to an open chamber and employ micropipettes to quickly probe a few vesicles before time is up (ie tension is decaying). Then we must start over with a freshly re-processed batch of vesicles. In new experiments, we did this for 8 batches of vesicles, now included in Figure 6. A rather boring result, for vesicles of 20 μm diameter or less, having compact domains, or star patterns, was that we could not detect an elevated membrane tension at all. This is expected since these vesicles are hypothesized to have low tensions and any tension would decay rapidly in the smaller vesicles. For vesicles exceeding 20 μm diameter, containing flowers, we did measure finite tensions, but with lysis expected near 6-8 mN/m , tension was high but variable, consistent with burst-reseal. By including these results, we hope the manuscript is made more compelling and hopefully do not distract with the complexities. Of note the high tensions measured do fall within the expected lysis range.

2. If direct determination of tension is completely impossible, the proposed steady state dependence of the tension (Eq. 1) should at least, be tested by direct experiments, carried out for a range of dT/dt values (rather than just two examples). Particularly informative would be the behavior at $dT/dt \rightarrow 0$ (quasistatic regime): does the R-dependence of domain shapes disappear upon the thermal equilibration of the sample? While the fundamental theoretical model considers an equilibrium balance between bending and line tension, the experiments are only carried out at the steady state water leakage conditions. For the theoretical mechanism to be validated, it is crucial to repeat the experiments at (or near) the equilibrium conditions.

Au response: We did include the tension above. Additional experiments for a range of cooling rates make sense and two cooling rates are included in Figure 5, but delving into fine quantitation adds little to the take home message of the current paper. We do plan a systematic

study of cooling rate for a later publication. The ultra slow cooling history is a challenging experiential we are working on for a future paper.

3. The experimental cooling protocol (lines 354-357) is too complex. It involves heating for a short time to a high temperature, then cooling at one rate, then cooling at a different rate. Since the rate of water squeezing out of thermally-compressed vesicles controls their tension and, consequently, shapes the crystalline domains, a simpler protocol should be adopted: pre-equilibration at a high temperature, followed by cooling at just one, well-defined, rate.

Au response: The reviewer is right about the physics of what is happening, but wrong with the judgement on “too complex.” We settled on the two step protocol so we could equilibrate the vesicles in all runs at unified conditions in the one phase region before attempting various cooling trajectories. This works with our cooling bath configuration. We employed temperatures near 50C to melt the pure crystals from the stored suspensions but did not want to overheat for long, and so cool back down, still in the one phase region to equilibrate a bit closer to the two phase envelope. From there we launch into the two phase region at different targeted rates. The heating and first cooling step are always the same. It seems complex and maybe unnecessary for the runs we published here, but will likely be more important for other studies such as the ultra-slow cooling proposed by this reviewer in #2.

4. Figure 3 is supposed to demonstrate "a primarily flat or cylindrically curved solid domain shape" (lines 164-167). In addition, it should exhibit a developable folding, with the Gaussian curvature focused at generator intersections (lines 201-202). However, since the solid is not fluorescent and appears dark, one can only speculate about its exact shape and folding. A different method of visualization is needed here. Could bright field microscopy or a fluorescent staining of the aqueous phase help here?

Au response: We agree that some visualization of solid shape is important. We were not successful in finding a dye that intercalated into the solid, despite new tracers described above. We therefore conducted additional experiments, replacing Figure 3, comparing epifluorescent and phase contrast/bright field images to reveal the shapes of parts of solid domains at the vesicle equator. The epifluorescent and phase contrast/bright field images are placed side-by-side with schematics that clarify expectations for vesicle orientations. One must look closely to discern flat parts of the equatorial profile to identify zero-Gaussian curvature regions.

5. The theoretical model does not allow the crystalline domain to adopt any other shape, beyond a 6-fold symmetric flower shape with a variable petal to core aspect ratio (Eq. 3). Thus, highly anisotropic protrusions and stripes, observed in some previous works (Refs. 5-8) are excluded. Topological structure defects are excluded as well. Thus, the fact that the model "reproduces" the experimental flower shapes does not prove the validity of the theoretical model: the shapes are simply built in. Since the validation of the domain shaping mechanism is the main goal of the present manuscript,

and since the experimental R-dependence is complicated by the above-mentioned issues (due to the non-equilibrium aspects of the process), it could help a lot if the authors show that the fundamental equilibrium theoretical considerations do indeed lead to the formation of 6-fold symmetric flower shapes, without these shapes being built "by hand" into the model.

Au Response: We agree with the basic point of the reviewer, that we do not attempt to model the origin of 6-fold symmetry of the solid domains. Although we disagree with the notion that the main point of the manuscript is to confirm the model. Instead, our presentation intends to present a surprising set of observations that run counter to existing physical mechanisms and ask instead if we can develop a minimal theoretical understanding of the underlying physics and mechanics that give rise to these unexplained, but clearly systematic and strong interdependence of emergent domain shape and vesicle size. We return to the point of topological defects in the responses below, but our basic assumption is that the predominant 6-fold symmetry of solid domains, whether compact or non-convex flowers), a reflection anisotropic line tension associated with the rotational symmetry of the molecular packing of solid. Hence, the faceted (hexagonal) shapes for compact domains are expected to result from standard considerations of equilibrium crystal shape (i.e. Wulff construction). As elastic energy favors more elaborate shapes on more inflated domains, as is evidently from the experimental results, this 6-fold symmetry is preserved, which is arguably evidence that the rotation symmetry of the molecular packing is largely preserved over the multi-micron domains in both compact and flower shaped solid domains. This suggests indirectly that defect formation that disrupts rotational symmetry, such as disclinations or polydomain grain boundaries, are not playing a significant role.

Based on this comment, and other reviewer suggestions, we recognize that this assumption about the role of anisotropic line tension setting the 6-fold symmetry in the experimental system is not clearly described in the manuscript. We have revised the introduction to the theoretical model to include a brief description of this underlying assumption on page 12.

Of course, the reviewer is correct to point out that one could construct and explore a more elaborate model that incorporates a new variable describing the anisotropy of line tension in order to test its specific role in the specific selection of 6-fold symmetry over other possible non-convex shapes (say, with 4-, 5-, 7-fold symmetry). We have no way to modulate this effect in experiments, or even characterize its quantitative value. The present study first aims to address the more basic question about what experimentally controllable processes select between convex and non-convex shapes. Indeed, ongoing theoretical work is exploring a more general model to explain the emergence of symmetry-breaking in elastic ground states for "isotropic" solid 2D domains.

Minor issues and suggestions

=====

6. The manuscript should discuss the reason for the absence of flower shapes in other works, where crystals are similarly frustrated by the Gaussian curvature (e.g., Refs 5-8). This may have to do with the curvature being tunable for the presently-studied vesicles,

while the curvature is fixed in other works, where e.g. colloidal particles reside on a droplet, the spherical shape of which is dictated by its relatively-high surface tension.

Au response: We agree that we can better emphasize the differences between the central mechanisms for morphology mechanisms previously studied, although we note that “flower-like” solid domains are predicted in Ref. 7, where the 6-fold symmetry is a reflection of the underlying crystalline packing in the model. The most important difference between those (and other prior studies in modeling and theory) is that the spherical template is held constant and inflexible. In contrast, vesicles are “topologically spherical” (i.e. there is a fixed average Gaussian curvature) but their shapes are flexible (i.e. Gaussian curvature can vary locally at finite elastic cost). This is critical, because it allows for low energy bending modes to expel Gaussian curvature from the solid in order to limit the expensive 2D elastic strain energy that is considered to drive shape transitions in references 5-8. We have revised the concluding paragraph of the article on page 22, to briefly reinforce this key distinction.

More surprising is the apparent absence of topological defects in the present system. The manuscript states that “the formation of topological defects that form in some curved 2D crystals[34, 35, 36] can only relax some, but not all, of the ... costs of Gaussian curvature.” This statement is not completely correct. Note that the strain of the perfect icosahedron-shape two-dimensional crystalline shell, with 12 disclinations residing at the vertices, is zero. Thus, one could imagine the growing developable crystalline domains in the presently-studied system to follow the shape of such a perfect icosahedral shell, incurring no strain-related penalties. This point should be discussed in the manuscript.

Au response: We should emphasize that can’t directly observe the absence or presence of topological defects in the solid phase of the experimental system. As above, we take the preserved 6-fold symmetry for nearly all observed solids as an indirect indication that the long-range rotational symmetry of the molecular packing in the solid is preserved in both compact and non-convex flower domains. This suggests that disclinations and tilt grain boundaries, which disrupt rotation symmetry of crystals, are not significant in the experimental system.

To be clear, the envisaged 5-fold defect of the reviewer does have a singular concentration of bending energy at the core of the defect, as well as the fold lines between defects. We are well aware of the elastic mechanisms of topological defect formation for spherical crystals (see e.g. DOI: 10.1103/PhysRevLett.112.225502 and 10.1103/PhysRevE.94.013003) and can therefore offer fairly generic scaling reasons why defect formation is a much more elastically expensive than simply redistributing Gaussian curvature from fluid to the solid. The elastic energy of forming a single disclination in a disclike solid is $\sim B \ln(\frac{W}{t})$ where we have assumed the limit of thin sheets proposed by the reviewer and the solid buckles into a conical shape with Gaussian curvature focused at the defect. Here, $t \equiv \sqrt{\frac{B}{Y}}$ is the elastic thickness of the membrane and $W \gg t$ is the multi-micron width of the solid and the logarithmic dependence on $\frac{W}{t}$ derives from the residual strain and singular bending at the defect core. Notably, in order to flatten the solid

domain in a flexible vesicle, it only requires redistributing some the Gaussian curvature of the solid portion to the fluid, increasing its only the *bending energy* of the fluid. As shown in the results of Fig. 4, this costs only an amount of increased bending energy $\sim B$ relative to a homogeneous vesicle. Since $\frac{W}{t} \sim 10^4 - 10^5$ our expectation is that disclination formation is significantly more expensive than simply absorbing the necessary Gaussian curvature in the fluid. Indeed, from the images in Fig. 4, it is clear that Gaussian curvature tends to concentrate regions at the periphery of the solid in the fluid domain, achieving the effect of disclinations in terms of Gaussian curvature concentration, but without the singular cost of strain concentration.

A more advanced theoretical model, called for in #5 (above), may allow the conditions for icosahedral, highly anisotropic stripe-like, and flower-shape domains to be delineated.

Au Reponse: Certainly a more complex model could be developed (and studied outside of the context of surface evolver) and studied to understand under what conditions, if any, defect formation might preempt redistribution of Gaussian curvature from solid to fluid domains in the vesicle. But, for the reasons emphasized above, we'd expect that to be relevant potentially to the regime when $W \approx t$, i.e. for nanoscale vesicles of radii closer to the nanoscopic thickness.

7. The theoretical alpha values, shown as a color map in Figure 5, exhibit no dependence on R when the reduced vesicle size is $\gg 1$. This is hardly surprising, as $R/(B \sigma^{-1})=1$ is the crossover between the bending- and the line tension-dominated regimes. I suggest that the authors consider changing the range of R-axis accordingly: one can either use a logarithmic R scale or introduce an axis break, allowing the high-R range to be compressed.

Au response: The reviewer is correct that there is a large region of state space for vesicles of small reduced volume where compact ($\alpha \sim 1$) is preferred, which is the expected (and intuitive) result. The regime where various flowers (elevated alpha) occurs when v_r is small requires a very very small line tension, which is away from what is experimentally accessible. So yes, in reality and for quite a substantial range in line tensions, compact domains are expected for even modestly deflated vesicles. That said, we appreciate the potential interest in this other super-low line tension regime and have included an inset containing the log scale version of the figure.

8. The relatively big gap in the experimental data in Figure 5 ($0.973 < \bar{v} < 0.984$) should be filled or explained, if possible. The gap is even larger for the squares. Is it a result of having the experimental data somehow thresholded?

Au response: The gap to which the reviewer refers falls between vesicles having hexagonal domains and star domains. It is quite tricky to distinguish a perfect hexagon from a shape starting to form a star (having a concave section of its sides) and once this is decided, the central portion of the solid is then identified and measured and the shape treated as hex or star. This along with measurement error contributes to the gap. Since the measurement error scales as pixel size

relative to vesicle size, it is greater for small vesicles (having compact hex/star) domains. Representative error bars have been added.

9. The difference between the 20 degC/hour data and the 70 degC/hour data in Figure 5 is difficult to see. Consider adding a trend line or changing the R range to stronger emphasize the difference between these data sets.

Au response: We appreciate the interest in adding a trend line, however, we believe a trend line implies a trend or concept that we do not intend to convey. We do not wish to imply that there is a predicted mathematical relationship describing the points measured at a given cooling rate. Such a relation may exist, integrating the non-equilibrium and non-linear processes of cooling dependence but no such master relation exists at the stage of this work. Here we simply employ different cooling rates to access different regions of state space, specifically different degrees of inflation/tension on vesicles of different sizes. Changing the cooling rate allows us to access a broader swath of state space, our only point here.

10. Why is the Poisson ratio taken equal to 0.4? What happens for other values?

Au response: Our expectation that the 2D Poisson ratio of the solid phase plays little to no role in the results presented. This is because, as shown in a new SI Fig. S14, the elastic strain energy in the solid phases is negligible relative to the bending cost over the relevant range reduced volume, since the solid has expelled the source of in-plane strain (Gaussian curvature) to the fluid phase. We chose the value of 0.4 following one reference 51 which attempted to estimate this parameter computationally .

11. What breaks the 6-fold rotation symmetry of H and KG curvature distributions (Figure 4A) for $\alpha \geq 1.1$? The solid is 6-fold symmetric, so that it is surprising that H and KG are not symmetric under horizontal rotation about the center of the solid domain. Remarkably, the $\alpha=1$ distributions, where the solid is circular, seem to exhibit a 4-fold symmetric H and KG distributions. The origin of these symmetry breaking effects should be discussed in the manuscript.

Au response: We agree with the reviewer that this is an interesting question, and one that we don't entirely have a complete answer to yet. Our heuristic understanding from the results presented in Fig. 4, is that the elastic cost of solid domain on a solid/fluid composite is largely controlled by the size of the planar core at its center. As shown clearly in the inset of Fig. 4C, the smaller the effective core, the lower the elastic energy. Hence, the elastic ground states tend to "squeeze this region down" to a size that is even smaller than the actual core. This is accomplished by introducing internal "cylindrical folds" that extend in straight lines through the solid from one boundary to the other. We posit that favorable fold patterns inscribe the smallest possible polygon extending from the solid edges. For the cases of the $\alpha = 1.1 - 1.4$ that would evidently be triangular pattern of folds that achieves a smaller core than is possible than the hexagonal pattern.

We are currently exploring this question of symmetry breaking in more depth in the context of axisymmetric solid domains, but following the reviewer's suggestion have added a sentence to the discussion of Fig. 4 results to summarize our current heuristic understanding.

12. The generators in Figure S9 (dashed lines) end, suspiciously, precisely at the sawtooth-like discontinuities which originate in the finite mesh size of Surface Evolver simulations. I wonder, whether these generators stay at the same positions or even survive, if a finer mesh is adopted for the simulations.

Au response: In our originally submitted manuscript, we simply estimated the positions of the generators based on vertical height information. Following the reviewer's question, we have developed a discrete differential geometry analysis of the surface evolver meshes to reconstruct the entire curvature tensor on the vesicle and render the generators directly from the stream lines of the principle curvature directions (i.e. following the directions of the locally flat direction). In Fig. S11 we have now render the exactly computed results of this reconstruction. The location of the generators in the image along the petals is arbitrary and not locked to the mesh.

13. The cartoon in Fig. 1A is not clear. The cut in the bilayer has a different orientation in the main panel and in the inset. The 3D perspective is incorrect, so that the cartoon's meaning is difficult to understand.

Au response: We have changed the illustration to increase clarity.

14. It is not clear, what are: "slightly inflated conditions (i.e. \bar{v} above the minimal elastic energy state)" (Line 186)? The value of \bar{v} used in the calculation should be specified.

Au response: We have revised Fig. 4B to indicate the values of the \bar{v} plotted in Fig. 4A and its location relative to the zero tension state (i.e. where E_{bar} is minimum).

15. The calculation of the tie line should either be completely removed to the Supplemental Information, together with Figure 1B (!), or explained more clearly in the main text. The present explanation is only good for the experts, not for the general Nat. Commun. readers.

Au response: The phase diagram is important to include so that readers understand the main features of the system: that the crystals are nearly the pure and that the fluid is a mixture. The tie line is a horizontal line across the phase diagram at room temperature and is not calculated. Apparently the original wording, referring to a calculation in the SI for the expected area fractions from the tie line was confusing. This language is updated for clarity on page 4, referring to the calculation in the SI.

16. I recommend the authors to consider completely removing the discussion of the effective core model to the Supplemental Information. The universal behaviour of $l(\tau)$, while interesting for itself, contributes very little to the main topic of the paper.

Au response: As described above, the value of the effective core model is two-fold. First, it allows us to test our heuristic idea that the elastic energy of fluid/solid vesicles is the effectively controlled by the planar core. Second, it allows us to present the essential connection between inflation and membrane tension, that is the critical link between vesicle size and solid domain shape.

17. Why is the $E(\text{strain})$, in Line 150, proportional to ϕ^3 ? The dependence on ϕ does not seem to be important in what follows. Therefore, I recommend the authors to choose whether to explain it properly or remove it (keeping just the R -dependence).

Au response: We have followed the reviewer's suggestion and removed the dependence on ϕ in these scaling analyses. The ϕ^3 dependence comes from energy of spherical cap of width W , see e.g. DOI: 10.1103/PhysRevE.94.013003. The elastic strain grows at $(W/R)^2$ and hence the elastic energy grows a $Y W^6 / R^4$. Since W^2/R^2 is proportional to ϕ , you have the result quoted in the text.

18. I recommend that Line 182, explaining the dimensionless measure of inflation, comes before the words "under nearly inextensible conditions...", to help the reader.

Au response: We have made this suggested revision.

19. Line 180: "with a minimum at a particular inflation ... that decreases over a series of solid shapes with increasingly larger petals (increasing α)." This statement should be clarified. The minimum gets deeper with α . The v_0 value INCREASES with α . It is not clear, what quantity does "decrease" for increasing α .

Au response: We have revised this description in accordance with the reviewer's suggestion.

20. Line 32 in the Supplemental Information: using R for two different quantities is somewhat misleading.

Au response: We have made a revision to address this issue, assigning "c" as the ratio of the two molar areas in the first section of the SI.

21. I also spotted several typos:

Line 213: "corresponding tensed"

Line 241: "conform to more uniformly to"

Line 398: "defined it perimeter"

Line 402: "adjusted to fixed the solid"

Line 416: "where σ " and P "are the line energy and perimeter"

Au response: Thanks, we have fixed these.

Reviewer 2B? No reviewer listed here.

The manuscript by Wan et al., on “Flowering of Developable 2D Crystal Shapes in Closed, Fluid Membranes” presents valuable insights into how the morphologies of 2D crystals growing in an elastic 2D fluid are controlled by a fundamentally new mechanism. There are some exciting new observations: the authors demonstrate a strong correlation between domain shape and vesicle size, while they use the same unique chemical composition of vesicles throughout the experiments. Furthermore, they present a detailed energetic model that elucidates the vesicle size-based selectivity of crystal morphology. The manuscript is well written, and the authors have conducted commendable work which may have a significant importance for related fields in science.

Discussed below are several things that require further attention to improve this version of manuscript:

Methodology and Data:

The study design and data analysis seem robust and appropriate for the research questions addressed. I'd like to suggest testing another set of chemical composition of the fluid membrane to ensure the claimed conclusion (i.e., vesicle size selects crystal shape). Although the data contains over 330 vesicles (manuscript line 117), validating the mechanism with another lipid composition (either a different bulk concentration of DPPC or a different pair of lipids) would strengthen the validity of the findings. If there is a particular reason for the specific DPPC concentration (30%), this point should be clearly explained in the text.

Au response: The reason for 30% DPPC was to produce one crystal per vesicle. This point is now amplified in the text on page 4-5. To address reviewer concern we conducted additional experiments at 60% DPPC/40% DOPC, but would like to provide them only the rebuttal, as they introduce expected complexities: Overloading the membrane with DPPC has several complicating effects but does not change our conclusions. First greater DPPC content increases the number of nuclei per vesicle (see Chen, Santore Langmuir 2014), so that multiple domains nucleate and interact during growth, changing their shapes. The petals can never grow so long as seen in Figure 1 because of this redistribution of growing solid material. Also with more DPPC, there is an important contraction of the membrane from the loss of fluid region and production of denser solid. This happens when there is 13 % solid too, but it is a small effect with less DPPC. With greater DPPC content, there is a greater membrane contraction, elevating tension and favoring flowers more. All that said, with a shifting of the distribution of shapes and poorly formed domains due to domain interactions, we still see more compact domains on small vesicles and the formation of petals on the domains in big vesicles, at included in this response to reviewer document.

Significance:

The paper addresses an important issue in the field of curved two-dimensional crystals, but it has a lack in explaining the potential uses and applications of the phenomenon

under investigation. Providing a comprehensive discussion on the potential real-world implications and practical applications of the discovered phenomenon would significantly enhance the manuscript's value to the Nature Communications' readers and further emphasize its significance for the broader scientific community. In addition, I strongly recommend inclusion of a short paragraph that elaborates on the importance of this phenomenon for other related fields of science. This could provide valuable insights and further enrich the discussion of tangent fields in the manuscript.

Au response: Please find a discussion of significance at the end of the outlook section, just before Methods.

Simulations:

Based on my understanding of the theoretical model, it appears that the presence of the 6-fold symmetric flower domain shape is predetermined and fixed within the model. Unfortunately, the model does not consider the influence of topological defects or allow the domain to adopt a preferred shape based solely on energetic considerations. As a result, the model is unable to provide the sufficient support for the discussed approach mechanism.

Au response: As described in detail in response to reviewer 2, we expect the 6-fold symmetry emerges in the experimental system due to the line tension of the crystalline domain, and therefore make no particular attempt to model its origin. Again, we can't directly see whether defects exist in the solid since the molecular order is invisible to experimental probes. We can only offer the fact that six-fold symmetry is apparently *not disrupted* as indirect evidence that disclinations and grain boundaries are not forming to neutralized the effect of Gaussian curvature (as suggested by the reviewer), as well as some basic scaling arguments to argue that for multi-micron vesicles redistribution of Gaussian curvature from the solid to the fluid is likely to be significantly lower elastic energy than cost of defect incorporation. Furthermore, the fact that our theoretical mechanism captures the observed experimental correlations *without* the inclusion of topological defects is consistent with the interpretation that they play little role in the process domain shape selection observed here.

Additional comments:

The figures and graphs included are informative and have detailed captions, but I suggest a slight change in Fig 5 since it's hard to tell the difference between the two cooling rates: triangles and squares. Trend lines might help.

Au response: We played with different symbols and colors but find that the symbols in the original paper stand out and distinguish the two cooling rates best. Adding a trend line suggests a functional dependence that we do not wish to imply in this paper. The aim of including two cooling rates was to argue for generality and to expand the region of state space we could access.

Overall, the manuscript is organized logically and coherently, and I find it interesting and valuable. I strongly recommend it to be accepted, given that the authors will take my comments into consideration.

Reviewer #3 (Remarks to the Author):

I co-reviewed this manuscript with one of the reviewers who provided the listed reports as part of the Nature Communications initiative to facilitate training in peer review and appropriate recognition for co-reviewers.

Reviewer #4 (Remarks to the Author):

The manuscript reports the results of a detailed experimental study of the morphologies of two-dimensional (2D) crystals, nucleated, grown, and integrated within the fluid membrane of giant vesicle. The shape of the crystal domains is found to be dictated by the interplay of mechanics, membrane permeability, and thermal vesicle contraction. The cost of solid strain of crystalline domains with Gaussian curvature drives formation of crystals with locally planar or cylindrical shapes, which expels Gaussian curvature into fluid membrane regions. Upon cooling to grow the crystals, large vesicles are found to sustain greater inflation and tension, which leads to the formation of more elaborate shapes on large vesicles despite their more gradual curvature, while small vesicles display compact planar crystals. This size dependence is opposite to the behavior of crystals on rigid spherical surfaces.

The interplay of curvature elasticity of membranes, the in-plane stretching elasticity of crystalline, the domain formation in multicomponent systems, and the membrane deformation in three dimensions gives rise to a multitude of interesting and often unexpected properties of vesicles. In the current manuscript, it is the expulsion of Gaussian curvature from crystalline domains coupled with the induced curvature of the embedding fluid membrane which generates intriguing flower-like domain shapes, which differ strongly from those on spherical geometries. Thus, the current study significantly enhances the understanding of complex membrane behaviors.

The following questions and comments should be addressed:

(1) The authors emphasize that "... morphologies of 2D crystals growing within an elastic 2D fluid having a flexible shape and a closed topology ... are controlled by a fundamentally distinct mechanism (than for fixed spherical templates)". Similarly, they state: "Remarkably, the most compact shapes are found on the small vesicles, i.e. those with the greatest curvature, while non-convex flowers are seen on larger vesicles, having smaller curvature. This correlation runs counter to ... that show the threshold size for the transition from compact to anisotropic domain shape increases with larger sphere radius".

I think that in this form and generality, these statements are somewhat misleading. The reason is that IN THE DESCRIBED EXPERIMENTS, it is not only the membrane curvature and deformability which determines the observed shapes, but equally important the thermal membrane contractions and water permeation, which generates a size-dependent membrane tension. This should be discussed more transparently.

Au response: we do feel that we have been transparent as thermal contractions and membrane permeability are prioritized in text:

- The first sentence of the abstract says, "The morphologies of 2D crystals nucleated, grown, and integrated within 2D elastic fluids, for instance giant unilamellar vesicle membranes, are dictated by an interplay of mechanics, permeability, and thermal contraction."

- Above, the reviewers first quote comes from last paragraph of the introduction. After that statement the paragraph explains how the mechanics of bending play out for the crystals. But then, in the same paragraph, it is noted that the size dependence of the morphological instability runs counter to crystallization on rigid spherical templates. We then say that thermal membrane contractions and water permeation are the critical variables that produce this size dependence, and at scale!

We therefore believe we are being transparent.

In the additional quotation provided by the reviewer, “Remarkably, the most compact shapes are found on the small vesicles, i.e. those with the greatest curvature, while non-convex flowers are seen on larger vesicles, having smaller curvature. This correlation runs counter to that show the threshold size for the transition from compact to anisotropic domain shape increases with larger sphere radius” The text is taken from the results section right at the point where the data on the size dependence are first introduced and so the discussion is focusing just on this data. Specific results later on directly address the mechanism for this size dependence: the combined impact of tension and permeability. Further we have added new tension data to argue for this part of the physics. We therefore believe we have not been misleading at any point.

(2) What is the role of line tension? In the current system, the line tension is clearly very small. For larger line tension, compact domains or even domain budding should be expected, see, e.g.

R. Lipowsky, J. Physique II, 2, 1825 1992);

R.J. Mashl et al., Biophys. J. 74, 2862 (1998);

T. Kohyama et al., Phys. Rev. E 68, 061905 (2003).

Au response: The impact of line tension was shown in Figure 5 of the original ms, on the y-axis. We highlight in pink where the original text mentioned line tension. The figure and discussion explain that line tension favors compact crystals and the line energy must be overcome for flower formation. This only happens when bending becomes expensive. The y-axis of Figure 5 shows the preferred shapes for different relative amounts of bending and line tension, with the x-axis addressing the vesicle inflation variable. So yes, as the reviewer points out, we do include that line tension favors compact domains. However, budding would not be expected in this work because the shear rigidity of the solid prevents the cap-shapes that form buds, even at relatively high line tensions. (Budding would be expected in vesicles having phase separation into two types of fluid domains, but that is different from the current fluid-solid vesicles.) We believe, therefore, that the ms has covered these points raised by the reviewer.

(3) In the spherical limit (of reduced volume approaching unity, or large tension), previous results for spherical surfaces should be recovered. From the results for spherical surfaces, I would expect petals to become stripe solid domains. Thus, it seems to me that petals and stripe solid domains are actually closely related polymorphs.

Au response: The reviewer has some ideas with which we agree: from the mechanical perspective, both stripes (in other works) and flowers petals (here) are developable, ie can bend in a cylinder shape maintaining zero Gaussian curvature around the vesicle. As a vesicle becomes more inflated, stripes (in other systems) and flowers (in our system) are preferred over compact solid domains. However, a perfect sphere is not achieved.

Regarding the issue of polymorphs: Are the petals and stripes related polymorphs? The DPPC stripes, seen in other works but not here, are the gel phase (Lb') and they are distinguished by being able to intercalate Rh-DPPE tracer lipid.(Chen, Santore PNAS 2014) Our compact domains and flowers (both core and petals) exclude this tracer and therefore are a different polymorph than the stripes of other works. In fact they exclude all tracers we have considered and we believe they are the Pb' (ripple) polymorph. In the main paper, we avoid the discussion of phospholipid solid polymorphs as they are somewhat specific to the particular lipid. In this paper, focusing on curvature, mechanics, etc, we imply we are dealing with just one kind of crystal packing, just one polymorph. In order to reach the broad audience and not only those with interest in lipids, we do not introduce the discussion of polymorphs, which would detract from our point, since we are working with just one polymorph (evidenced in the new data by uniform dye exclusion from the solid). Additionally the theory, which includes only one set of mechanical parameters for the solid, as opposed to one set of parameters for the core and a different set of moduli in the petals, predicts the experimental selectivity for compact or flowered domains, arguing that our observations could be expected for a single solid polymorph.

(4) The effective core model seems to imply that petals deformation energy becomes negligible in the spherical limit (high tension). Doesn't this again indicate the close relation of petals and stripe solid domains on spherical geometries?

Au response: As we said in (3), striped solid domains are a shape that could bend around a vesicle and maintain zero Gaussian curvature in the solid. And our petals are related to stripes in this way. But why a molecule or colloid packs in a crystal that is fundamentally stripey, versus fundamentally symmetric (6 fold here) has to do with the molecular packing, which is a detail we take as a given here. If we had a lipid that could only ever form stripes and did not prefer a 6-fold symmetric crystal (when flat), then it would not be fun to study the impact of curvature, we suppose.

(5) Two related phenomena, which the authors might want to briefly discuss, are:

-- Grain boundary scars in curved surfaces with crystalline order:

** rigid spherical shape: A.R. Bausch et al., Science 299, 1716 (2003);

** deformable vesicles: T. Kohyama et al., Phys. Rev. Lett. 98, 198101 (2007).

-- Multicomponent crystalline vesicles with different mechanical properties of the components:

** M.D. Emanuel et al., Phys. Rev. E 102, 062104 (2020);

** S. Li et al., ACS Nano 15, 14804 (2021).

Au response: We don't believe effects of grain boundary scars to be playing a particularly important role here, because our solid phase is *not* required to have Gaussian curvature, unlike

the case of the closed 2D spherical crystals studied in the references mentioned here. Again, the key difference is that the global topology of the vesicle requires there to be non-zero Gaussian curvature *somewhere* in the vesicle, but in the fluid/solid composite, it is possible (and energetically preferable) to concentrate it in the fluid phase, thereby avoiding the in-plane elastic costs of crystalline membranes forces to have spherical curvature.

Reviewer #5 (Remarks to the Author):

The manuscript considers growing crystals on flexible membranes and claims possibilities to control size and shape by the size of the membrane and growth conditions. The observed phenomena are nicely explained by reviewing theoretical results ranging from crystal growth on spherical surfaces, to elastic bending and stretching theory and their relation to intrinsic and extrinsic curvature. The experimental setup somehow can be seen as a realization of these theories and in this clearness I'm not aware of any other such study. I only have some concerns with the considered simulations. They are based on Surface Evolver and the considered setting is probably the best one can do within this software. However, the authors should at least mention and discuss the limitations.

1) The considered 6-fold symmetry is an input, the model does not contain such information. In [6,7] this symmetry results from the microscopic lattice structure. In case there is a possibility to measure this experimentally, it would be nice to make a statement on possible defects in the solid phase, or stress distributions of these solids to confirm the relation to the theories [6,7].

Au response: We don't know of a way to measure stress distribution in the solid. Such a method does not exist. We can infer some aspects of the solid shape. We have developed a new figure 3 which is the best we can do to elucidate aspects of the solid shape, beyond what is inferred from the shape of the fluid phase. Because we don't see dye intercalating into the solid, we cannot see any defects if they occur. The idea is that with near zero Gaussian curvature and developable shapes, such defects are not likely to occur. See the responses to reviewer 2 for more detailed arguments about the relative elastic cost of defect formation vs. expelling Gaussian curvature from solid to fluid domain.

2) The relaxation, seems not to consider any relaxation in the phases, e.g. there is no line tension considered. While it is argued that the magnitude is much lower, a simultaneous relaxation of the phases and the shape leads to various additional coupling terms, which should not be ignored, see e.g. Bachini et al. arXiv:2305.15147 for a recent discussion in the context of two fluid phases.

Au response: We do take line tension into account. It is the σ on the y-axis of Figure 5. The relaxation determines the shape of the contour of the fluid and solid domains once the shape of the solid (α) is specified. The relaxation is executed independently for vesicles of different α and \bar{v} values and the resulting energies and vesicle shapes (given the shape of the solid domain "cut out") are shown in Figure 3. That includes the bending energy. However the line energy is a line tension times the solid perimeter and is taken into account in the state space of Figure 5, which shows how line tension versus bending energy shifts the preference between compact and flower shaped domain for different \bar{v} (inflation) values.

3) It should be explicitly stated that the simulations do not consider growth or any dynamic evolution.

Au response: Page 12 and the caption of Figure 3 explain the results are for a fixed area fraction and we therefore are not tracking the growth process. In order to model growth, in principal we could carry out a series of surface evolver at intermediate domain size . We have slightly revised the text to make more clear that we are not attempting to capture this dynamical evolution.

4) It is argued that the bending rigidity is the set to be the same for solid and fluid phase for simplicity and the flattening of the solid phase is associated with the considered plate theory. However, also different bending energies already could lead to this effect, see again Bachini et al. arXiv:2305.15147. The plate theory would not be needed in this case. This should be discussed. I guess it could also be simulated within the Surface Evolver.

Au response: This statement is not quite true. The author suggests that we'd capture similar effects with simply a large contrast in the bending moduli, as opposed to the geometrically non-linear elastic cost of shear in plate theory. The effect of very large bending modulus in the solid phase (ignoring the artificial nature of this approximation) would be to promote *globally planar* (i.e. zero mean AND Gaussian curvature) solid domains, as opposed to developable shapes, which have zero Gaussian curvature and allow for mean curvature bending at an energy scale comparable to fluid bending. This latter scenario is much more clearly supported by experimental observation, as there is no question that the petals of the solid domain are bent with comparable curvature radius as the surrounding fluid. Nevertheless, the question about the different emergent shapes and morphologies is an interesting one and we hope our work inspires future work in this direction.

5) [4,7] should not be called phase-field modeling , but phase-field-crystal modeling, as the microscopic origin is essential.

Au response: Thanks. We have updated this.

1. Chen, D., Santore, M.M. Large effect of membrane tension on the fluid-solid phase transitions of two-component phosphatidylcholine vesicles. *Proceedings of the National Academy of Sciences of the United States of America* **111**, 179-184 (2014).
2. Bagatolli, L.A., Gratton, E. A correlation between lipid domain shape and binary phospholipid mixture composition in free standing bilayers: A two-photon fluorescence microscopy study. *Biophysical Journal* **79**, 434-447 (2000).
3. Chen, D., Santore, M.M. Three dimensional (temperature-tension-composition) phase map of mixed DOPC-DPPC vesicles: Two solid phases and a fluid phase coexist on three intersecting planes. *Biochimica Et Biophysica Acta-Biomembranes* **1838**, 2788-2797 (2014).
4. Chen, D., Santore, M.M. 1,2-Dipalmitoyl-sn-glycero-3-phosphocholine (DPPC)-Rich Domain Formation in Binary Phospholipid Vesicle Membranes: TwoDimensional Nucleation and Growth. *Langmuir* **30**, 9484-9493 (2014).
5. Witten, T.A. Stress focusing in elastic sheets. *Reviews of Modern Physics* **79**, 643-675 (2007).
6. Seung, H.S., Nelson, D.R. Defects in Flexible Membranes wiht Crystalline Order. *Physical Review A* **38**, 1005-1018 (1988).
7. Xin, W.Y., Wu, H., Grason, G.M., Santore, M.M. Switchable positioning of plate-like inclusions in lipid membranes: Elastically mediated interactions of planar colloids in 2D fluids. *Science Advances* **7**, eabf1943 (2021).

REVIEWER COMMENTS

Reviewer #1 (Remarks to the Author):

This reviewer carefully read the revised manuscript and the responses of the authors.

The reviewer appreciates the effort of the authors to conduct additional experiments as suggested both by this reviewer and by other reviewers as well.

In some instances, the experiments validated the almost "pure" "solid" packing of lipids within the domains. In other cases, the experimental design resulted in more than one domain per vesicle, not supporting the findings described in the original submission that was/is based on another, single, composition of lipids.

This reviewer appreciates the detailed revision but, in their mind, the original question remains; this is an interesting study that describes a "general mechanism and not just lipids"(*), to quote the author's statement. However, even the revised manuscript is still based on a single lipid composition. For any journal, it would be difficult to accept a manuscript claiming a "general mechanism" based on a single composition only.

(* response to question '5' of this reviewer's comments in the first rebuttal: "The suggestion to study different multiple lipid systems does not add to the understanding of new mechanism in this work, which targets a general mechanism and not just lipids.")

In addition, the request to hold off on data (that are "off topic") is awkward. But this reviewer leaves this in the discretion of the editor.

Reviewer #2 (Remarks to the Author):

Reviewer #2

The present version of the manuscript, with the membrane tension measurements included, is an improvement over the previous version. Yet, some of the authors' responses to the issues raised in the previous round are still unsatisfactory, as detailed below. Furthermore, the new portions of the text are in part very difficult to understand, mainly due to a high concentration of grammatical errors. Since final decision must be taken in this round, **I recommend the manuscript to be accepted for publication**, despite the mentioned deficiencies. However, I strongly recommend that the authors introduce the textual revisions which are necessary to satisfactorily respond to the following comments.

1. To my question (in the previous report) regarding the breaking of the 6-fold symmetry of H and KG distributions, the authors respond that the energy is further reduced by squeezing the core along straight folding lines, which extend from one boundary of the solid to the other. According to this argument, the 6-fold symmetry is broken spontaneously by a triangular pattern of folds. This pattern of folds could adopt one of two different orientations, with respect to the 6-fold symmetric core. Clearly, the energy of either of these orientations is the same. Since H and KG distributions in Fig. 4A are produced by zero-temperature energy minimization calculations, it is not clear, how does the calculation choose one of the two possible broken symmetry orientations. In an experiment, such a choice necessitates the presence of noise, either a thermal or a non-thermal. What is the origin of noise in a zero-temperature numerical calculation? Is it a numerical noise? Is a different orientation (out of the two equal-energy options) achieved upon repeating the same calculation? This point must be clarified in the manuscript.

2. In my previous report, I expressed a concern as for the 6-fold symmetry being introduced “by hand” into the theoretical model. The authors chose to avoid constructing a more elaborate model, motivating their decision by the fact that they “have no way to modulate this effect in experiments”. However, it must be noted, that in the limit of a negligible line tension, the current model is invalid. In particular, the 6-fold symmetric petals are unstable to splitting into a larger number of narrower petals. Each finite-width petal on a sphere experiences a finite Gaussian curvature, which may be reduced by this petal’s splitting in two narrower petals. Even if the Gaussian curvature is completely expelled from the petals, the concentration of the mean curvature energy at the petals’ edges would still be decreased by petals’ splitting. In an absence of line tension, there is nothing to oppose such a splitting, so that the validity of the 6-fold symmetric model is questionable. The authors should clarify this issue in the manuscript.

3. I do not understand the authors’ claim (in their Rebuttal letter) that the ultra-slow cooling rates are challenging to study experimentally. This point should be clearly explained in the manuscript, to avoid the readers being puzzled by the absence of equilibrium results from the story. Since the effect should disappear for an ultra-slow cooling, such an experiment is, in my opinion, the most direct test of the proposed scenario. I fail to imagine a controlled-cooling setup which DOES allow a fast cooling (at several different rates!), yet DOES NOT allow for a slow cooling.

4. There are many sentences which must be revised by a native English speaker, prior to publication. Following are a few examples:

(A) Caption of Fig. 4 “is chosen to match the tension-free states for both models.” Should it be: “chosen to have the tension-free state E -bar match for both of the models”?

(B) Lines 254-255: “...effective core model, which effectively assumes the dominant elastic (bending) costs to derive the edge of a nearly planar region at the core of the solid.” - This sentence is not clear. Should it be “to determine the edge”?

(C) Line 256: “In effective core model, elastic of effect of the solid domain is modeled...” - Correct this text.

(D) Line 389: “which makes the incorporate of Gaussian...” - correct the grammar.

(E) Line 300-302: “Notably, given the estimated values of line tension, the Figure 5 inset highlights that dependence of the of increased line tension on non-compact shapes (...) is only expected to becomes strong relative to its dependence on inflation for vesicles of one or two orders of magnitude smaller radius.” - In addition to the grammatical errors, the sentence is far too long and does not make sense. The line tension is a constant parameter. What does it mean that the line tension depends on non-compact shapes?! Revise this sentence entirely.

Note, there are many such sentences in the text and all of them must be corrected, not just the few examples which are listed above.

5. The manuscript states (Line 232): “This folding is evident in the images in Figure 3C between the petals in real vesicles.” It is not clear, what feature of the images in Fig. 3C do the authors refer to. Either explain it in more detail or (better) add small arrows to the figure, pointing to the relevant features.

6. In their Rebuttal letter, the authors state that the (partial) relaxation of Gaussian curvature by disclinations is only possible for $W \approx t$, i.e. for nanoscale vesicles of radii close to the nanoscopic thickness. However, the regime discussed in References 38-39, which have been mentioned in my report (these are References 35-36 of the earlier version of the manuscript) is the same as in the current work: $W \gg t$. Yet, the disclinations play an important role in relaxing the Gaussian curvature in these studies. Apparently, the difference is in the value of the line tension, rather than in the W/t ratio.

7. I did not point out this issue in Round 1, but it should be noted that the value assumed (line 499) for the line tension (1 kT/nm) seems to be just a wild guess. The reported values of lipids’ line tension

vary by ~ 5 orders of magnitude [see Stottrup et al., Langmuir 35, 16053 (2019)]. The text in the manuscript should be explicit on this issue.

As for the authors' request to remove Figures RR-1 and RR-2 from the rebuttal letter, I agree that Figure RR-2 can be removed. However, since Figure RR-1 is the only demonstration that crystal shape selection rules work for a single-step cooling, I recommend this figure to be introduced into the Supplemental Information file.

Reviewer #2B

In the revised manuscript the authors have demonstrated a commendable effort in addressing my comments from my initial review, and the manuscript now is much improved and emphasizes important points for clarity. I strongly recommend this manuscript to be accepted.

As for the figures appearing in the rebuttal letter:

I agree with the authors, that the new set of experiments (Fig RR-2) is not necessary to convey the main message of the manuscript. Thus, it may be removed from the published rebuttal, as the authors suggest. The revised manuscript provides a more detailed rationale for selecting the lipid composition (30:70) and points out the complexities in choosing other compositions. Additionally, the experiments' data could be beneficial for future publications, allowing the authors to go deeper into the additional physical mechanisms that they mentioned.

In the revised manuscript the authors added a good, satisfying description of the theoretical model, and why this model is convincing enough in order to understand the domain shape selection. I agree that in the current state of things, the topological defects play a minor role in this process. Therefore, the existing theoretical model is adequate for this purpose.

Reviewer #3 (Remarks to the Author):

Reviewer #4 (Remarks to the Author):

In their reply letter, the authors have responded to all points of my previous report in some detail. Although this has not led to any significant changes in the manuscript, I recommend the publication of the manuscript in its present form.

Reviewer #5 (Remarks to the Author):

Most of my questions have been answered and I support publication after some minor changes.

- I think on l36 from the cited refs it is not clear what is the effect of Gaussian curvature, as these refs consider stationary surfaces

- at l71, models for fluid membranes also consider Gaussian curvature terms, either within the Helfrich energy, or more prominently if surface viscosity is considered in the context of fluid deformable surfaces, see e.g. Torres-Sances et al. JFM 872 (2019) 218; Reuther et al. JFM 900 (2020) R8 and Krause et al. JCP 486 (2023) 112097.

- at l118, please justify the choice of the same bending rigidity for both phases, at least flat solid parts can also be achieved by larger rigidity in the solid phase
- l206, it is argued that the 6-fold symmetry results from an anisotropic line-energy, this is probably correct, however, in the provided simulation it results as the shape is specified having a 6-fold symmetry, this should be explicitly mentioned
- l460, Gaussian bending contributions are neglected, the considered argument only holds if the Gaussian bending rigidity is constant in both phases, the arguments on the difference with respect to the Gaussian curvature in the solid and fluid phase suggest a different rigidity, please explain and justify this approach
- reduce the number of typos and add all the missing information in the references

We are again grateful to all the reviewers for their careful reading and thoughtful input on this work.

Response to Reviewer Comments on R1

REVIEWER COMMENTS

Reviewer #1 (Remarks to the Author):

This reviewer carefully read the revised manuscript and the responses of the authors.

The reviewer appreciates the effort of the authors to conduct additional experiments as suggested both by this reviewer and by other reviewers as well.

In some instances, the experiments validated the almost “pure” “solid” packing of lipids within the domains. In other cases, the experimental design resulted in more than one domain per vesicle, not supporting the findings described in the original submission that was/is based on another, single, composition of lipids.

This reviewer appreciates the detailed revision but, in their mind, the original question remains; this is an interesting study that describes a “general mechanism and not just lipids”(*), to quote the author's statement. However, even the revised manuscript is still based on a single lipid composition. For any journal, it would be difficult to accept a manuscript claiming a “general mechanism” based on a single composition only.

(* response to question ‘5’ of this reviewer’s comments in the first rebuttal: “The suggestion to study different multiple lipid systems does not add to the understanding of new mechanism in this work, which targets a general mechanism and not just lipids.”)

In addition, the request to hold off on data (that are “off topic”) is awkward. But this reviewer leaves this in the discretion of the editor.

Authors: This paper, which combines experiments on phospholipid vesicles with a continuum model, presents a general mechanism for solid (crystal) morphologies in curved 2D fluids. We emphasize that the model is a continuum treatment that is not dependent on the choice of molecule, and that it broadly predicts a class of behavior. It also agrees with and explains a mechanism for the experiments. The generality of the model, not being restricted to any particular kind of molecule is an important point and so we have emphasized it in the second revision in the last sentence of the introduction. The paper’s establishing of a general mechanism is therefore based on much more than a single lipid composition: The paper combines a lipid composition with a theory that makes no restrictions at the molecular level.

The paper focuses on conditions in which a single solid domain crystallizes within each vesicle to make the behavior easy to see and to quantify, a point we clarified in the first revision and which was noted by the reviewer 2B.

In our response to the first round comments by reviewer 1 and to a request by the editor for additional experiments specifically at a composition that was a majority DPPC, we provided images from further experiments, conducted with vesicles of 60% DPPC/40% DOPC. In that case, now, DPPC was not only the majority component, it was also the one that crystallized. The selection of 60% DPPC/40% DOPC therefore created the situation of greater numbers of DPPC solid domains on each vesicle (as expected

from literature) and greater solid area fraction on each vesicle (necessitated by the phase diagram). While having larger / multiple domains per vesicle added complexity, it was seen in the images of our first response that the key observation was preserved even in vesicles with multiple domains: Flower domains or domains with concave regions were seen on larger vesicles and compact convex domains on the smaller vesicles. We therefore disagree with the statement of reviewer 1, “not supporting the findings described in the original submission”. The data we provided in our first response absolutely did support the original findings.

Reviewer 1 further challenges us that the paper itself should include multiple compositions to be convincing about the generality of the mechanism. We continue our assertion that such an exercise is outside the scope of this work. This paper quantitatively addresses a new mechanism based on experiments that closely compare with a continuum model, addressing the energies of fluid and solid phases with minimal constraints on the properties of either, and not requiring one type or another of molecular interaction. The data we provide in the main paper, at two separate cooling rates employ vesicles whose morphologies and tensions are quantified and compared to theory with great care. A sampling across a broad range of compositions, and quantitative study of the impact of different solid areas distributed across multiple domains is a separate topic for another paper.

For editorial review, to convince the editors that mechanism we reported is indeed a general one, we provide, for review only, new data for 20% DPPC/80 % DOPC, in this response. Also substantiating the generality of our results but for review only are new data at 30 wt% DPPC / 70 wt% commercial copolymer Xiameter OFX-5329 (a lamella former), nominally the same copolymer employed in Chen and Santore *Soft Matter* 2015. This commercial copolymer, producing many multi-lamellar (very bright) vesicles in addition to some unilamellar vesicles, shows the trend of flower-shaped domains on large unilamellar vesicles and compact domains on small ones.

[redacted]

Reviewer #2

The present version of the manuscript, with the membrane tension measurements included, is an improvement over the previous version. Yet, some of the authors' responses to the issues raised in the previous round are still unsatisfactory, as detailed below. Furthermore, the new portions of the text are in part very difficult to understand, mainly due to a high concentration of grammatical errors. Since final decision must be taken in this round, **I recommend the manuscript to be accepted for publication**, despite the mentioned deficiencies. However, I strongly recommend that the authors introduce the textual revisions which are necessary to satisfactorily respond to the following comments.

1. To my question (in the previous report) regarding the breaking of the 6-fold symmetry of H and KG distributions, the authors respond that the energy is further reduced by squeezing the core along straight folding lines, which extend from one boundary of the solid to the other. According to this argument, the 6-fold symmetry is broken spontaneously by a triangular pattern of folds. This pattern of folds could adopt one of two different orientations, with respect to the 6-fold symmetric core. Clearly, the energy of either of these orientations is the same. Since H and KG distributions in Fig. 4A are produced by zero-temperature energy minimization calculations, it is not clear, how does the calculation choose one of the two possible broken symmetry orientations. In an experiment, such a choice necessitates the presence of noise, either a thermal or a non-thermal. What is the origin of noise in a zero-temperature numerical calculation? Is it a numerical noise? Is a different orientation (out of the two equal-energy options) achieved upon repeating the same calculation? This point must be clarified in the manuscript.

Authors: We thank the reviewer for raising this clear question. There are two sources of “symmetry-breaking noise” in the calculations. The first comes from the discretization effects associated with finite-elements used by Surface Evolver. The triangulation of the original spherical mesh already breaks symmetry, but more significant, we expect, are commensurability effects between the solid-fluid boundary, which we define via a smooth curve and the underlying triangular mesh, which lead to slight roughening of the solid edge on a scale of the triangular mesh. We expect this effect is sufficient to perturb the symmetry to bias the internal polygonal folding to one of possibly many nearly degenerate states. Beyond this, the numerical minimization routine incorporates a random “jiggle” displacement of vertices to check for the stability of putative local minima (as summarized in the SI section of the surface evolver protocol). This would also have the effect of introducing symmetry-breaking perturbations. We have revised the manuscript on the bottom of page 13 and top of page 14 to briefly note the intrinsic sources of “symmetry-breaking noise” in our computations.

2. In my previous report, I expressed a concern as for the 6-fold symmetry being introduced “by hand” into the theoretical model. The authors chose to avoid constructing a more elaborate model, motivating their decision by the fact that they “have no way to modulate this effect in experiments”. However, it must be noted, that in the limit of a negligible line tension, the current model is invalid. In particular, the 6-fold symmetric petals are unstable to splitting into a larger number of narrower petals. Each finite-width petal on a sphere experiences a finite Gaussian curvature, which may be reduced by this petal's splitting in two narrower petals. Even if the Gaussian curvature is completely expelled from the petals, the concentration of the mean curvature energy at the petals' edges would still be decreased by petals' splitting. In an absence of line tension, there is nothing to oppose such a splitting, so that the validity of the 6-fold symmetric model is questionable. The authors should clarify this issue in the manuscript.

Authors: We are in no way claiming that surface tension can be ignored. To be clear, we are not including *anisotropy* of the solid-fluid boundary in our numerical model, although it is the likely source

of preserved 6-fold symmetry. In 2D crystals, one can understand the line energy as periodic function of the domain edge orientation relative to a local crystallographic symmetry axis (e.g. for hexagonal crystals, it is 6-fold symmetric function of local orientation). In the present manuscript, model considers a non-zero line energy (see the y-axis of Fig. 5), but neglects its modulation with local angle. To capture the emergent faceted shape via standard consideration (i.e. the Wulff construction, [https://doi.org/10.1016/0370-1573\(84\)90066-8](https://doi.org/10.1016/0370-1573(84)90066-8)) one includes not only the average value of line energy, but also its periodic modulation (i.e. the higher modes in the Fourier series). We certainly agree that in the absence of such an orientational field to favor 6-fold symmetry, the emergent symmetry of the non-convex shapes driven by a preference for minimal bending energy in the fluid-(developable) solid composite is not determined. Although we feel this is something of an artificial question (i.e. motivated by the model rather than the observed physical phenomenon), it is an interesting one from the perspective of the model itself that we hope to address in future research. Our expectation is that without a specific molecular field directing the symmetry, the emergent symmetry (say n-fold symmetric flower-like domains) of favored domains will exhibit a complex dependence on the inflation, solid fraction and (appropriately dimensionless) ratio of line tension to bending energy. We have revised the manuscript on page 12 to better highlight this point.

3. I do not understand the authors' claim (in their Rebuttal letter) that the ultra-slow cooling rates are challenging to study experimentally. This point should be clearly explained in the manuscript, to avoid the readers being puzzled by the absence of equilibrium results from the story. Since the effect should disappear for an ultra-slow cooling, such an experiment is, in my opinion, the most direct test of the proposed scenario. I fail to imagine a controlled-cooling setup which DOES allow a fast cooling (at several different rates!), yet DOES NOT allow for a slow cooling.

Authors: Cooling and heating rate experiments are typically challenging. Ultraslow ramps, especially cooling are still more challenging. We designed a thermal ramp arrangement for the particular experiments we wished to conduct and which appear in the main paper. It turns out to be difficult to work much outside the targeted range. Worth noting, we developed a custom arrangement because a system that meets our needs is not commercially available. Temperature control stages for microscope slides introduce thermal gradients and primarily control heating, though a cooling add-on with an external bath can be purchased. Ours is a microscope slide arrangement that holds the liquid vesicle suspension, taped to a heating-cooling manifold through which a temperature control fluid circulates from a bath. It is not a heating coil on a feedback controller, such found in a DSC. In the manuscript, the main data are acquired for 0.3 C/min. Targeting ultraslow cooling of 0.8 C/ hour we require, 0.013 C/minute, ramping for over 15 hours. We have not been able to do this with our set up. For reference please note that the rate of 0.01 C/min corresponds to the slowest possible rate on a commercial DSC that employs feedback control on a ~10 mg sample in a thermally conductive aluminum pan. Our specimens are larger and the glass is a poorer conductor than aluminum. It would be particularly challenging to design a system that would resolve temperature measurements to a fraction of a degree and impose a feedback loop with sufficiently tight control to avoid overshooting and cycling. The nature of such real experimental challenges which are common to a broad swath of the soft material field, seem inappropriate to include in the experimental section.

4. There are many sentences which must be revised by a native English speaker, prior to publication.

Following are a few examples:

Authors: The actual challenges lie in describing the relationship between multiple variables.

(A) Caption of Fig. 4 “is chosen to match the tension-free states for both models.” Should it be: “chosen to have the tension-free state $E\text{-bar}$ match for both of the models”?

Authors: We have changed the language to clarify that $E\text{-bar}$ is the variable that is matching in the two models. Thanks.

(B) Lines 254-255: “...effective core model, which effectively assumes the dominant elastic (bending) costs to derive the edge of a nearly planar region at the core of the solid.” - This sentence is not clear. Should it be “to determine the edge”?

Authors: We have attempted to stream line this and other parts of this part of the writing.

(C) Line 256: “In effective core model, elastic of effect of the solid domain is modeled...” – Correct this text.

Authors: We have attempted to stream line this and other parts of this part of the writing.

(D) Line 389: “which makes the incorporate of Gaussian...” – correct the grammar.

Authors: Done, thanks.

(E) Line 300-302: “Notably, given the estimated values of line tension, the Figure 5 inset highlights that dependence of the of increased line tension on non-compact shapes (...) is only expected to becomes strong relative to its dependence on inflation for vesicles of one or two orders of magnitude smaller radius.” – In addition to the grammatical errors, the sentence is far too long and does not make sense. The line tension is a constant parameter. What does it mean that the line tension depends on non-compact shapes?! Revise this sentence entirely.

Authors: While line tension is fixed in an experiment, is a variable in the state space of Figure 5 and this writing was attempting to address when the morphology is sensitive to the choice of line tension and when it is not. This section has been rewritten.

Note, there are many such sentences in the text and all of them must be corrected, not just the few examples which are listed above.

Authors: We did read through everything again and tried to catch, especially, ambiguities in descriptions of relationships between variables.

5. The manuscript states (Line 232): “This folding is evident in the images in Figure 3C between the petals in real vesicles.” It is not clear, what feature of the images in Fig. 3C do the authors refer to. Either explain it in more detail or (better) add small arrows to the figure, pointing to the relevant features.

Authors: We have updated the text to read:

“The focused Gaussian curvature is evident in the images in Figure 3C between the petals in real vesicles, especially in the phase contrast image where the cross section appears to have corners in the fluid phase.”

6. In their Rebuttal letter, the authors state that the (partial) relaxation of Gaussian curvature by disclinations is only possible for $W \approx t$, i.e. for nanoscale vesicles of radii close to the nanoscopic thickness. However, the regime discussed in References 38-39, which have been mentioned in my report (these are References 35-36 of the earlier version of the manuscript) is the same as in the current work: $W \gg t$. Yet, the disclinations play an important role in relaxing the Gaussian curvature in these studies. Apparently, the difference is in the value of the line tension, rather than in the W/t

ratio.

Authors: Critical and distinct assumptions of the models in ref. 38-39 and the present work are that the 2D crystal is perfectly closed and enveloping the entire drop (i.e. $\Phi = 1$). In such a case, at least a minimal number of (12) disclination defects are required by *topology* rather than energetics. In our current case, the solid domains cover only a small ($\Phi \ll 1$) fraction of the vesicle surface. The reviewer should note (as described in ref. 41) that the even when imposing a rigidly spherical shape on the solid domain, a single disclination is only favorable when the solid exceeds 1/6 of the surface area. We have not yet studied this interesting question in full detail, but for the reasons summarized previously, our expectation is this threshold is push to even higher values if shape is allowed relax on flexible vesicle (depending, of course, on its dimensionless volume), not to mention relaxing the shape of the fluid-solid domain boundary.

7. I did not point out this issue in Round 1, but it should be noted that the value assumed (line 499) for the line tension (1 kT/nm) seems to be just a wild guess. The reported values of lipids' line tension vary by ~5 orders of magnitude [see Stottrup et al., Langmuir 35, 16053 (2019)]. The text in the manuscript should be explicit on this issue.

Authors: Our estimate of the range of this value is based on the values and ranges as cited earlier and now appearing as references 47 and 48 in this revision. Certainly we concede that the value of this parameter in the particular regime of fluid-solid domain coexistences is especially poorly characterized. We note only our physical model (as summarized in Fig. 5) suggests that the boundary between compact and flower domains is especially *insensitive* to changes of this parameter (over orders of magnitude). We have revised our manuscript on page 18 to point out the ambiguity in this physical parameter and its importance (or lack thereof) for the conclusions presented in this study.

As for the authors' request to remove Figures RR-1 and RR-2 from the rebuttal letter, I agree that Figure RR-2 can be removed. However, since Figure RR-1 is the only demonstration that crystal shape selection rules work for a single-step cooling, I recommend this figure to be introduced into the Supplemental Information file.

Authors: We have revised the supporting information accordingly and mention it in the experimental section.

Reviewer #2B

In the revised manuscript the authors have demonstrated a commendable effort in addressing my comments from my initial review, and the manuscript now is much improved and emphasizes important points for clarity. I strongly recommend this manuscript to be accepted.

As for the figures appearing in the rebuttal letter:

I agree with the authors, that the new set of experiments (Fig RR-2) is not necessary to convey the main message of the manuscript. Thus, it may be removed from the published rebuttal, as the authors suggest. The revised manuscript provides a more detailed rationale for selecting the lipid composition (30:70) and points out the complexities in choosing other compositions. Additionally, the experiments' data could be beneficial for future publications, allowing the authors to go deeper into the additional physical mechanisms that they mentioned.

In the revised manuscript the authors added a good, satisfying description of the theoretical model, and why this model is convincing enough in order to understand the domain shape selection. I agree that in the current state of things, the topological defects play a minor role in this process. Therefore, the existing theoretical model is adequate for this purpose.

Reviewer #3

Reviewer #4

In their reply letter, the authors have responded to all points of my previous report in some detail. Although this has not led to any significant changes in the manuscript, I recommend the publication of the manuscript in its present form.

Reviewer #5

Most of my questions have been answered and I support publication after some minor changes.

- I think on l36 from the cited refs it is not clear what is the effect of Gaussian curvature, as these refs consider stationary surfaces

Authors: As described in these references, the effect of Gaussian curvature (i.e. fixed spherical shape) is to frustrate the (strain-free) equal neighbor spacing in crystalline assemblies, which gives rise to states of internal stress in crystalline domains (with spherical curvature) that grows rapidly with lateral dimension. Based on this effect, models in these references predict that it is favorable for assembling crystals on *fixed spheres* to limit their lateral domain sizes, lowering the elastic cost of strain energy at the expense of increasing line energy.

- at l71, models for fluid membranes also consider Gaussian curvature terms, either within the Helfrich energy, or more prominently if surface viscosity is considered in the context of fluid deformable surfaces, see e.g. Torres-Sances et al. JFM 872 (2019) 218; Reuther et al. JFM 900 (2020) R8 and Krause et al. JCP 486 (2023) 112097.

Authors: For reasons detailed below (see response to comment about l460) the role of the linear coupling to Gaussian curvature plays no role in mechanisms studied in this manuscript. This is because the geometrically non-linear suppression of Gaussian curvature ***by the solid domain in-plane elasticity is much stronger in effect than the linear Gaussian curvature term in the Helfrich energy.*** The reason for this, is well described in the many references cited in this manuscript (e.g. refs. 5-8) but summarized succinctly by the scaling arguments on page 9, that estimate the ratio of in-plane strain costs to Helfrich bending terms, which is the same for the mean curvature (squared) and (linear) Gaussian curvature term. Moreover, as argued below, combining the vanishing of Gaussian curvature on the solid with the topologically-fixed integral of Gaussian curvature over the composite vesicle implies that the presence of linear Gaussian curvature coupling from the fluid domain has *no effect* on domain shape selection (i.e. its energetic contribution is *fixed for any solid domain shape of the same area fraction*).

- at l118, please justify the choice of the same bending rigidity for both phases, at least flat solid parts can also be achieved by larger rigidity in the solid phase

Authors: This is not quite true, since the solid domains are predicted and observed to bend significantly, with (mean) curvatures in the “petals” of the solid being comparable to that of the fluid. The key difference is that the strain energy of the solid leads to prohibitive elastic cost specifically for *Gaussian curvature*, but mean curvature leads to no in-plane solid strain and is only penalized by bending energy. We concede that the solid domain is likely to have at least a slightly higher bending modulus, but it is important to note that setting the bending stiffness of the solid domain arbitrarily larger than the fluid would not give rise to the same compact to non-convex/wrapped morphology transition observed in experiments. Instead, solid domains would simply be planar and there would be no thermodynamic incentive to form “cylindrically curved” petals which can better conform to a quasi-spherical shape. As our primary objective is to develop a mechanistic understanding of this otherwise mysterious morphology transition observed in fluid-solid composite vesicles, as at present bending modulus difference between fluid and solid phases in this system has not been quantified experimentally, we adopt the simplest assumption of equal bending modulus for the purposes of simplicity, as mentioned in the Methods section. The fact that the model predicts a preference for flowers with bendy petals to enable solid domains to fit on vesicles that are substantially inflated emphasizes that the mechanism does not require that the solid be more expensive to bend than the fluid. Instead, the key physics is the shear rigidity of the solid, distinguishing it from the fluid.

- I206, it is argued that the 6-fold symmetry results from an anisotropic line-energy, this is probably correct, however, in the provided simulation sit results as the shape is specified having a 6-fold symmetry, this should be explicitly mentioned

Authors: We attempted to make this point explicitly clear in our previous revision on page 12. Following this suggestion, and the suggestion of Reviewer 2, we have revised this discussion of the assumed origin of the 6-fold symmetry and the specific assumptions of the present model on page 12.

- I460, Gaussin bending contributions are neglected, the considered argument only holds if the Gaussian bending riidity is constant in both phases, the arguments on the difference with respect to the Gaussian curvature in the solid and fluid phase suggest a different rigidity, please explain and justify this approach

Authors: The negligibility of the effects of integrated Gaussian curvature terms in this study does not rely on having the same Gaussian curvature modulus. This derives from the combined facts that 1) Gaussian curvature in the solid domain is zero (due to the prohibit cost of shear elastic strain discussed in detail in this paper and as evidence in surface evolver results in Fig. 4) and 2) the surface tangent is continuous at the fluid-solid phase boundary. These combined ingredients imply that integrated Gaussian curvature over the fluid domain is always 4π , while it is obviously zero for the solid. This result can be understood from the fact that integral of Gaussian curvature over the composite vesicle is topologically invariant and equal 4π , while the contribution from the (isometrically flat) solid is zero. We have added a sentence to the methods section on p 26 to clarify this point.

- reduce the number of typos and add all the missing information in the references

Authors: we have done another read through and hopefully have not missed anything.

REVIEWERS' COMMENTS

Reviewer #1 (Remarks to the Author):

The new set of images that are provided by the authors on the composition 80%wt DOPC + 20% wt DPPC is the much needed proof, to this reviewer, that the morphological behavior described in the manuscript is true for at least a second composition of the same lipid mixture. This data/images validate the generality in the behavior claimed in the previous versions of the manuscript and should be included in the final version of the manuscript; not to "convince the editors", but to "convince" the readers/the scientific community.

Reviewer #2 (Remarks to the Author):

Reviewer #2

The present version of the manuscript responds to most of my concerns, so that I recommend it to be accepted for publication in Nat. Commun. Since the authors maintain that carrying a slower cooling experiment requires a significant effort in modifying the temperature control setup, I agree that delaying the publication of the present manuscript till after this construction is completed may possibly be unjustified.

Reviewer #3 (Remarks to the Author):

Reviewer #5 (Remarks to the Author):

The manuscript has further improved. Unfortunately the suggestion to discuss a potential impact of surface viscosity in the fluid phase has not been discussed in the manuscript. Also the added comments on Gaussian bending rigidity are only partly sufficient. The argumentation already assumes to have zero Gaussian curvature in the solid phases. It is no longer a result of the model. This should at least be mentioned. Several refs still do not contain any article number. I can recommend the paper to be published in NC but suggest to address the above points.

Response to Reviewers

Reviewer #1 (Remarks to the Author):

The new set of images that are provided by the authors on the composition 80%wt DOPC + 20% wt DPPC is the much needed proof, to this reviewer, that the morphological behavior described in the manuscript is true for at least a second composition of the same lipid mixture. This data/images validate the generality in the behavior claimed in the previous versions of the manuscript and should be included in the final version of the manuscript; not to “convince the editors”, but to “convince” the readers/the scientific community.

Au: These are now included in the Supplemental Information Figure 6

Reviewer #2 (Remarks to the Author):

Reviewer #2

The present version of the manuscript responds to most of my concerns, so that I recommend it to be accepted for publication in Nat. Commun. Since the authors maintain that carrying a slower cooling experiment requires a significant effort in modifying the temperature control setup, I agree that delaying the publication of the present manuscript till after this construction is completed may possibly be unjustified.

Au: Thank you!

Reviewer #3 (Remarks to the Author):

Au: Thank you!

Reviewer #5 (Remarks to the Author):

The manuscript has further improved. Unfortunately the suggestion to discuss a potential impact of surface viscodity in the fluid phase has not been discussed in the manuscript. Also the added comments on Gaussian bending rigidity are only partly sufficient. The argumentation already assumes to have zero Gaussian curvature in the solid phases. It is no longer a result of the model. This should at least be mentioned. Several refs still do not contain any article number. I can recommend the pater to be publishes in NC but suggest to address the above points.

Au: We now address this on page p26,27, of the main manuscript, explaining that while K_g is not strictly constrained to be zero for the solid domains, it turns out to be zero within resolvable limits. We then back this up in Supplementary Figure 8.